# Environmental damping and vibrational coupling of confined fluids within isolated carbon nanotubes

Yu-Ming Tu [1,11], Matthias Kuehne [1,2,11], Rahul Prasanna Misra [1], Cody L. Ritt[1], Hananeh Oliaei [3], Samuel Faucher[1], Haokun Li[4], Xintong Xu [4], Aubrey Penn[5], Sungyun Yang [1], Jing Fan Yang [1], Kyle Sendgikoski[6], Joshika Chakraverty[1], John Cumings [7], Arun Majumdar[4,8], Narayana R. Aluru [9], Jordan A. Hachtel [10], Daniel Blankschtein [1] & Michael S. Strano [1] ✉

Because of their large surface areas, nanotubes and nanowires demonstrate exquisite mechanical coupling to their surroundings, promising advanced sensors and nanomechanical devices. However, this environmental sensitivity has resulted in several ambiguous observations of vibrational coupling across various experiments. Herein, we demonstrate a temperature-dependent Radial Breathing Mode (RBM) frequency in free-standing, electron-diffraction-assigned Double-Walled Carbon Nanotubes (DWNTs) that shows an unexpected and thermally reversible frequency downshift of 10 to 15%, for systems isolated in vacuum. An analysis based on a harmonic oscillator model assigns the distinctive frequency cusp, produced over 93 scans of 3 distinct DWNTs, along with the hyperbolic trajectory, to a reversible increase in damping from graphitic ribbons on the exterior surface. Strain-dependent coupling from self-tensioned, suspended DWNTs maintains the ratio of spring-to-damping frequencies, producing a stable saturation of RBM in the low-tension limit. In contrast, when the interior of DWNTs is subjected to a water-filling process, the RBM thermal trajectory is altered to that of a Langmuir isobar and elliptical trajectories, allowing measurement of the enthalpy of confined fluid phase change. These mechanisms and quantitative theory provide new insights into the environmental coupling of nanomechanical systems and the implications for devices and nanofluidic conduits.

One-dimensional nanomaterials such as carbon nanotubes (CNTs) have exceptionally high surface areas per volume, enabling new types of sensors[1,2], energy storage devices[3], and nanomechanical and nanoelectromechanical systems[4,5]. There is significant interest in studying the thermodynamics and dynamics of fluids under extreme nano-confinement, where deviations from prevailing theory are appreciable. It is recognized that the interior of carbon nanotubes, as a nanofluidic conduit, offers a compelling platform for such study. However, understanding the mechanical and vibrational coupling of such systems to the surrounding environment, particularly with respect to resonances and vibrational modes, is critical for predicting device behavior. The most prominent vibrational mode for nanotube systems is the A1g axisymmetric expansion and contraction of the cylindrical shell, called the Radial Breathing Mode (RBM). This mode is widely utilized to approximately assign nanotubes due to its inverse dependence on diameter[6,7]. The surface area of the shell couples to mass in the environment,

inducing both frequency and scattering intensity changes in response, allowing researchers to study nanotube-nanotube interactions[8], lattice strain[8,9], the adsorption of condensed noble gases[10] and humidified air[11], fluidic filling in the interior[12–16], solution dispersion[17], and high-pressure diamond anvil cell (DAC) deformations[18–20].

Despite its widespread usage as an environmental sensor, a quantitative description of the RBM frequency with nanotube coupling across different environments has remained elusive. This has been exacerbated by seemingly contradictory observations across different experiments reported in the literature. Chiashi et al.[11] used the RBM to observe up to a 10 cm$^{-1}$ upshift upon exterior water adsorption to suspended CNTs when the humidity exceeded 7%. The RBM is also observed to upshift by as much as 17–24 cm$^{-1}$, which is of the order of 10%, upon CNT bundling, where van der Waals (vdW) contact results in the formation of parallel aggregates of CNTs, in contact[7,21]. These observations have been interpreted based on the harmonic model to correspond to an increase in the restoring force spring constant associated with the coupled mass, causing a distinct upshift in frequency. However, Kumar et al.[9], using strain on the PDMS substrate, pulled apart similar CNT bundles and observed an opposite 6 to 8 cm$^{-1}$ frequency upshift that the authors attribute to a change in damping on debundling. Similarly, it has been noted in past[12] and current experiments (Supplementary Text 1: a and Supplementary Fig. 1-1) that substrate-supported, un-opened CNTs show negligible or small (<1%) RBM upshifts when the outer surface is exposed to liquid water, despite having more than 80% of the external surface accessible, suggesting a highly non-additive dependence on coupled mass in the environment, which to date, has not been quantified.

Along these lines, we herein discover and analyze a system that should be an ideal control or reference state for an environmentally uncoupled CNT (Fig. 1). An unopened CNT suspended across transmission electron microscopy (TEM) windows removes the coupling from the substrate (Fig. 1a, b). Placed under vacuum, external fluids and the atmosphere are also removed as sources of coupling, especially after thermal annealing. Surprisingly, we show that approximately 2/3 of RBMs observed from preparation in this way exhibit a relatively large 9–14 cm$^{-1}$ reversible shift in the RBM upon laser heating, one of the largest ever observed for CNTs, and never before under vacuum (Fig. 1c–e). The remainder exhibits no measurable RBM frequency shift on heating (Supplementary Text 1: b and Supplementary Fig. 1-2), also indicating that the intrinsic temperature dependence of the RBM frequency is negligible, consistent with previous observations[22]. The magnitude of the 9–14 cm$^{-1}$ shift (>10 %) is relatively large compared with theoretical and experimental predictions for strain (<3%)[8], adsorbed water (<7 cm$^{-1}$, 4–7%)[11], or immersion in liquid Ar or Xe (<3 cm$^{-1}$ or 2%)[10].

Herein, we utilize this observation to develop the first quantitative model of RBM coupling that is able to describe these diverse environments. We show that it can describe this reversible temperature cycling over 93 scans of three electron-diffraction (ED) assigned Double-Walled Carbon Nanotubes (DWNTs), in addition to all other CNTs investigated using this system. This model is based on both the damping and restoring force that accompanies the coupling of mass to the nanotube shell. This spring-and-dashpot picture of mass coupling is able to resolve the apparent ambiguities of current and past observations. We show that upon filling the interior of the CNT with water, the trajectory markedly changes, as predicted by a harmonic oscillator force balance of the axial displacement. This new, quantitative understanding of environmental coupling enables a new generation of high-fidelity nanomechanical devices and fluid-sensing nanoconduits.

## Results
### Reversible Radial Breathing Mode thermal trajectories
The platform consists of an isolated, ultralong (mm) DWNT grown across 13 slits (most of which are 35 μm) of a 3 mm in diameter

homemade TEM chip such that micro-Raman spectroscopy can be performed on the suspended section in the absence of an underlying substrate (Fig. 1a, b)[23]. The lack of substrate and heat sink means that the laser fluence can be varied from $5.8 \times 10^3$ to $6.4 \times 10^5$ Watts/cm2, resulting in heating in the approximately 2 μm diameter spot from where the spectrum is collected from ambient to as high as 1,000 K. Unless otherwise specified, measurements were performed on a vacuum stage with pressure between $10^{-3}$ to $10^{-8}$ bar. The spot temperature is measured using the same CNT G-band and two-point calibration as in previous studies (Fig. 1c)[24,25]. The same system can then be imaged in TEM with ED assigning the chiral indices to the DWNT investigated. These DWNTs were selected for their resonant interior and exterior shells, both visible with a single excitation laser[26]. Throughout the heating cycle, the RBMs corresponding to the interior and exterior shells of the DWNT are visible in the resulting Raman spectrum (Fig. 1d), exhibiting a reversible sharpening and intensification at high temperature compared to ambient temperature. This study focuses on the systematic and reversible wavenumber shift, which demonstrates several unique features.

The system in Fig. 1a was intended as a control and starting point for experiments expected to induce environmental coupling and shift the RBM. Instead, even under a $10^{-8}$ bar vacuum, a large 9 to 14 cm$^{-1}$ (10 to 15%) shift of the low RBM frequency ($\omega_L$, frequency in-phase collective oscillation of the two carbon shells) is consistently observed upon heating, producing a distinctive trajectory on a subset of the samples. A (19,3)@(22,11) DWNT, referred to as CNT F, demonstrates this behavior in Fig. 1e. Here, the RBM shifts in a concave downward trajectory towards a cusp at a characteristic temperature, $T_{max}$. Subsequent heating ($T > T_{max}$) then keeps the RBM frequency invariant at $\omega_{RBM, min}$, as shown in Fig. 1e(i) at 565.4 K. Scan (i) is the first heating cycle, which most often results in a high temperature $T_{max}$ above 500 K, with all subsequent cycles on the same spot exhibiting a lower $T_{max}$ close in value as shown in Fig. 1e(ii),(iii), (355.8 and 352.8 K, respectively). The magnitude of the frequency shift is large, commensurate with the extreme deformations found in DAC experiments, where extreme radial deformations at high pressure are imposed[18–20]. The concave down trajectory towards the cusp at $T_{max}$ appears to be a distinctive feature of all CNTs, suspended and self-tensioned, exhibiting this behavior throughout 93 separate temperature scans (Supplementary Text 2–4 and Supplementary Fig. 3-1) in all three DWNT systems studied.

The only candidates for environmental coupling observed for these systems under vacuum with no underlying substrate are carbonaceous impurities, as shown in Fig. 1f. We assign them as having high aromatic content or graphitic domains, as they remain invariant even under repeated temperature cycling above 1000 K under pyrolysis conditions[27], known to increase aromatic content starting at 500 °C. These graphitic impurities were grown together with CNTs via chemical vapor deposition (CVD) synthesis after an oxidative cleaning step. We reason that these necessarily have graphitic domains in a ribbon-like structure, allowing them to bond strongly via pi-pi stacking, and remain adhered even with repeated temperature cycling above 700 K.

Several observations show clear ribbon-like structures, as in Fig. 1f(ii). From extensive TEM imaging, these entities are found to be ubiquitous on all CNTs in this study, including a statistical sampling of over 75 TEM imaged CNTs (Supplementary Text 5: a and Supplementary Fig. 5-1). A typical axial surface coverage of 60% is shown in Fig. 1f(vi). Using TEM videography, the diffusion of these ribbons from the Mean Squared Displacement (MSD) tracking is observed to be approximately 0.12 nm2 s$^{-1}$, translating to a time exceeding $10^6$ s to reversibly leave and enter the 2 μm laser spot (Supplementary Text 5: b, Supplementary Fig. 5-2, and Supplementary Movie 1). Consistent with this, the magnitude of the reversible RBM shift does not change upon 12 h of baking out under $10^{-5}$ bar vacuum at 100 °C. Hence, only

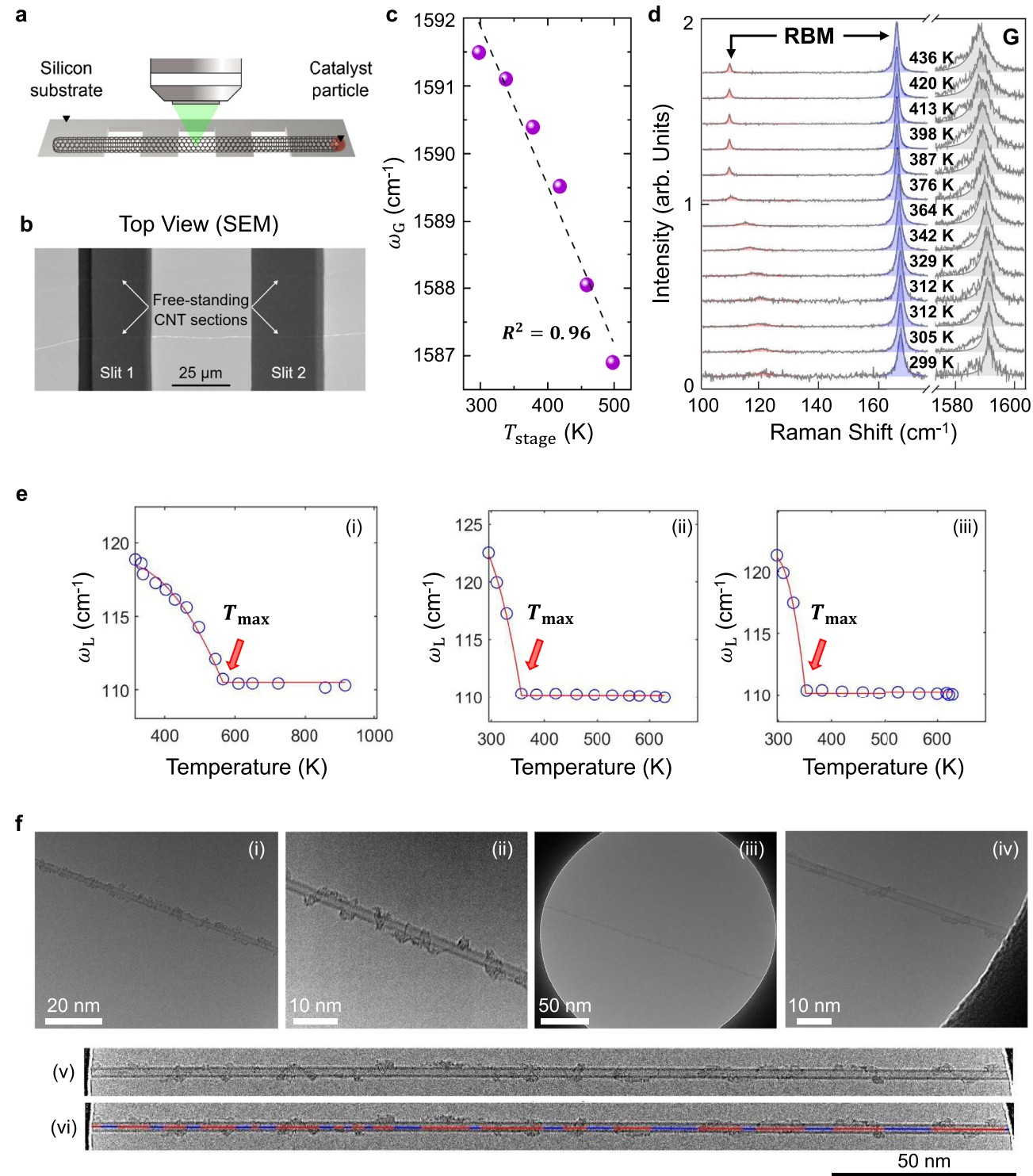

the relative strength of the graphitic coupling appears to be changing for $T<T_{max}$ and otherwise remains invariant at temperatures in excess of $T_{max}$.

**Reversible thermal trajectories of the RBM frequency for three assigned DWNTs**

These reversible thermal trajectories are further investigated using three DWNTs identified by TEM and lattice assigned by ED, labeled CNT G, CNT X, and CNT F. The heating/cooling direction is indicated by each arrow (Fig. 2). CNT G, a (29,1)@(35,6) DWNT by ED (Fig. 2a(i)), exhibits a large initial $T_{max}$ of 701.1 K at $1.2 \times 10^{-6}$ bar from room

temperature ($T_o$) (Fig. 2a(ii)). Subsequent heating and cooling scans (Fig. 2a(iii)–(vi)) show no other hysteresis, with $T_{max}$ values decreasing to between 423.1 and 400.0 K. CNT X is a (28,5)@(28,11) DWNT (Fig. 2b(i)) and does not exhibit the large $T_{max}$ when the initial scan starts at 445.7 K instead of $T_o$, indicating that this is needed to eliminate the initial scan hysteresis. Instead, a $T_{max}$ of 350.6 K is observed (Fig. 2b(ii)–(vi)). CNT F is a (19,3)@(22,11) DWNT and initially (Fig. 2c(ii)) exhibits a high $T_{max}$ of 565.4 K during heating from room temperature at $1.2 \times 10^{-6}$ bar. However, on repeating the heating and cooling scans, a consistent $T_{max}$ of around 350 K is observed (Fig. 2c(iii)–(v)). The trajectory ending at $T_{max}$ from experimental

**Fig. 1 | Platform and observations for the temperature-dependent Radial Breathing Mode (RBM) shift for isolated and suspended Double Walled Carbon Nanotubes (DWNTs). a** Schematic of isolated DWNT consisting of as-grown and free-standing entities. **b** Scanning electron microscopy (SEM) image of free-standing DWNTs. **c** Local CNT temperature calibration using stage temperature ($T_{stage}$) and the Raman G ($\omega_G$) responses with the linear fit (dash line). $R^2 = 0.96$. **d** Example Raman spectra of the as-grown, free-standing (19,3)@(22,11) DWNT as a function of local temperature heated by the excitation spot at 633 nm using a vacuum stage at $7.4 \times 10^{-7}$ bar. The Raman G-band was separately calibrated and used as a local thermometer of the CNT. Lorentzian fits (shaded curves) served to extract Raman mode frequencies. **e** Low frequency RBM ($\omega_L$) corresponding to the outer shell of the (19,3)@(22,11) DWNT (CNT F), demonstrating the characteristic trajectory observed for a subset of CNTs studied in this work. All trajectories consist of a concave down curve towards a cusp at a limiting temperature $T_{max}$, (red arrow) followed by an invariant RBM frequency for all $T > T_{max}$. Scan (i) is the initial heating

curve at $1.2 \times 10^{-6}$ bar towards 916.5 K, showing a $T_{max} = 565.4$ K. Subsequent scans on the same spot then show $T_{max}$ values consistently lower and repeatable. Scan (ii) heating at the same location towards 628.2 K with a $T_{max} = 355.8$ K and (iii) heating towards 621.5 K with $T_{max}$ at 352.8 K, both measured at $2.5 \times 10^{-3}$ bar. The red line is an analytical fit using Eq. (6), and the fitting parameters are provided in Supplementary Table 12-2. **f** Transmission electron microscopy (TEM) images of (i) a (15,6)@(20,12) DWNT with a 20 nm scale bar, (ii) sharper contrast at a 10 nm scale bar showing the specific structure. The graphitic impurities are found on every CNT imaged. A (21,6)@(24,14) DWNT (iii) with a 50 nm scale bar, (iv) with a 10 nm scale bar, showing similar graphitic and ribbon-like impurities. (v) A cropped TEM image from (iii) to show distinct (21,6)@(24,14) DWNT (Supplementary Movie 1). (vi) Segments that are labeled in red indicate the presence of graphitic carbon decoration at about 60% coverage per linear distance of the DWNT. The blue shaded lines are empty regions in Fig. 1f(v). Source data are provided as a Source Data File.

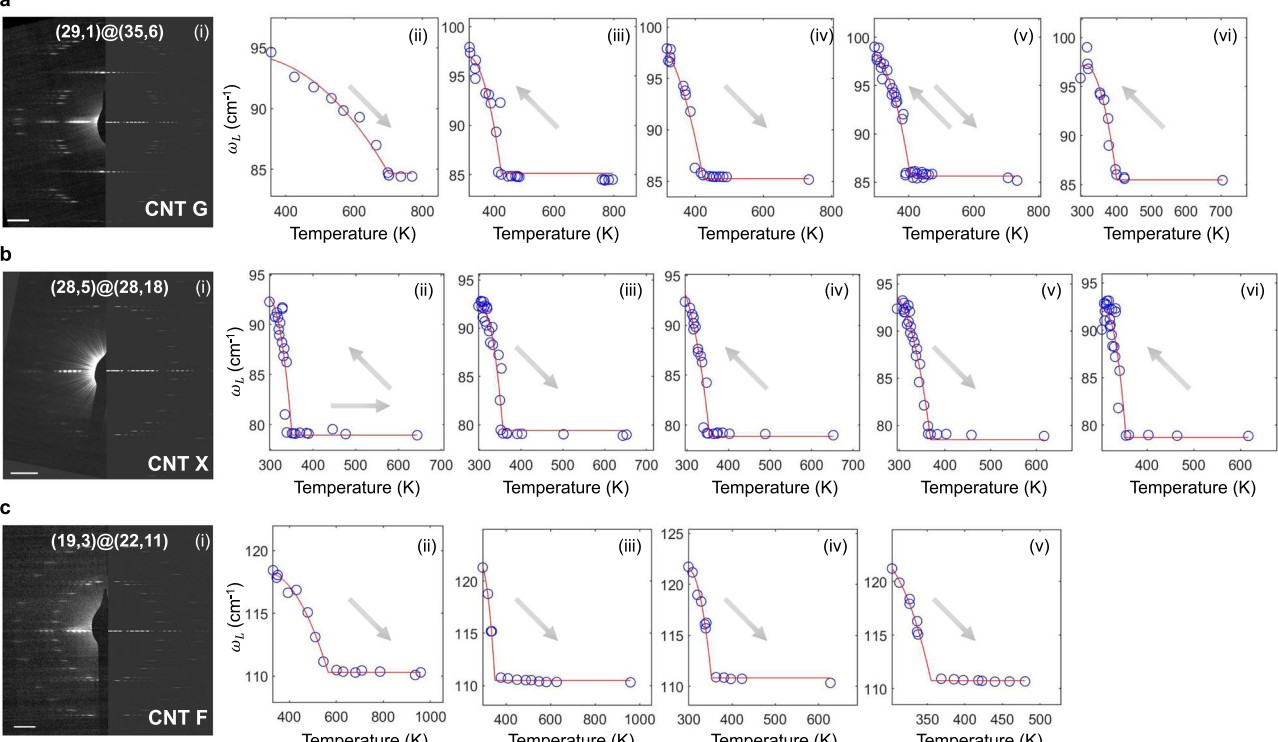

**Fig. 2 | Repeated heating and cooling scans for a series of DWNTs.** Arrows show the direction of temperature change during the scan. **a** CNT G is a (29,1)@(35,6) DWNT as identified by TEM and ED (left half: measured ED pattern, right half: simulated ED pattern using code DIFFRACT) (i). The initial heating at $1.2 \times 10^{-6}$ bar (ii) from room temperature ($T_o$) produces a large $T_{max} = 701.1$ K but subsequent cooling (iii), heating (iv), cooling then heating (v), and final cooling (vi) shows no other hysteresis. $T_{max}$ values from (ii) to (vi) are 423.1, 423.3, 405.0, and 400.0 K, respectively. **b** CNT X is a (28,5)@(28,11) DWNT by ED (i) and when the initial scan at $1.9 \times 10^{-8}$ bar starts at 445.7 K towards 643.8 K with subsequent cooling to $T_o$, the large $T_{max}$ is not observed, but rather 350.6 K for (ii), upon reheating (iii) yielding

356.5 K, cooling (iv) 354.4 K, heating (v) 368.7 K and final cooling (vi) 354.3 K. **c** CNT F is a (19,3)@(22,11) DWNT from ED (i) and heating from RT at $1.2 \times 10^{-6}$ bar (ii) shows the high $T_{max}$ at 565.4 K initially, but the value falls to a consistent 350 K upon heating again from $T_o$ (iii), tracing the same trajectory on repeating (iv) with $T_{max} = 352.3$ K and again (v) at 355.0 K. For all 93 scans analyzed in this work, the features of these trajectories are consistently observed. The red trace line in the plot is an analytical fit using Eq. (6), and the fitting parameters are provided in Supplementary Table 12. The scale bar in ED is 2 nm$^{-1}$. Source data are provided as a Source Data File.

observations (Figs. 1e and 2) suggests the scaling:

$$y[T] = \frac{T - T_o}{T_{max} - T_o} - 1 \quad (1)$$

where, the scaling factor $y[T]$ scales the transition of the damping term, $b$, with the temperature, $T$, according to the expression: $b = b_{max} + y[T]\Delta b$, where $\Delta b = b_{max} - b_{min}$ as well as $b_{max}$ and $b_{min}$ correspond to the damping at $T_{max}$ and $T_o$, respectively. The features of these RBM trajectories of an unopened and free-standing CNT system

are consistently observed in all 93 scans analyzed in this work (Supplementary Text 2–4).

**Environmental coupling through vibrational damping**

The RBM is described using a harmonic oscillator model with the radial force balance representing the radial displacement $w[t]$ containing terms accounting for the intrinsic CNT spring constant contribution ($\frac{\beta}{r^2}$), where $r$ is the CNT shell radius, and $\beta$ is the intrinsic spring constant, and environmental contribution ($\gamma$) (Fig. 3a)[13,18,28,29]. A damping term proportional to coefficient ($b = \frac{b}{2\rho}$) has been proposed[9] but not

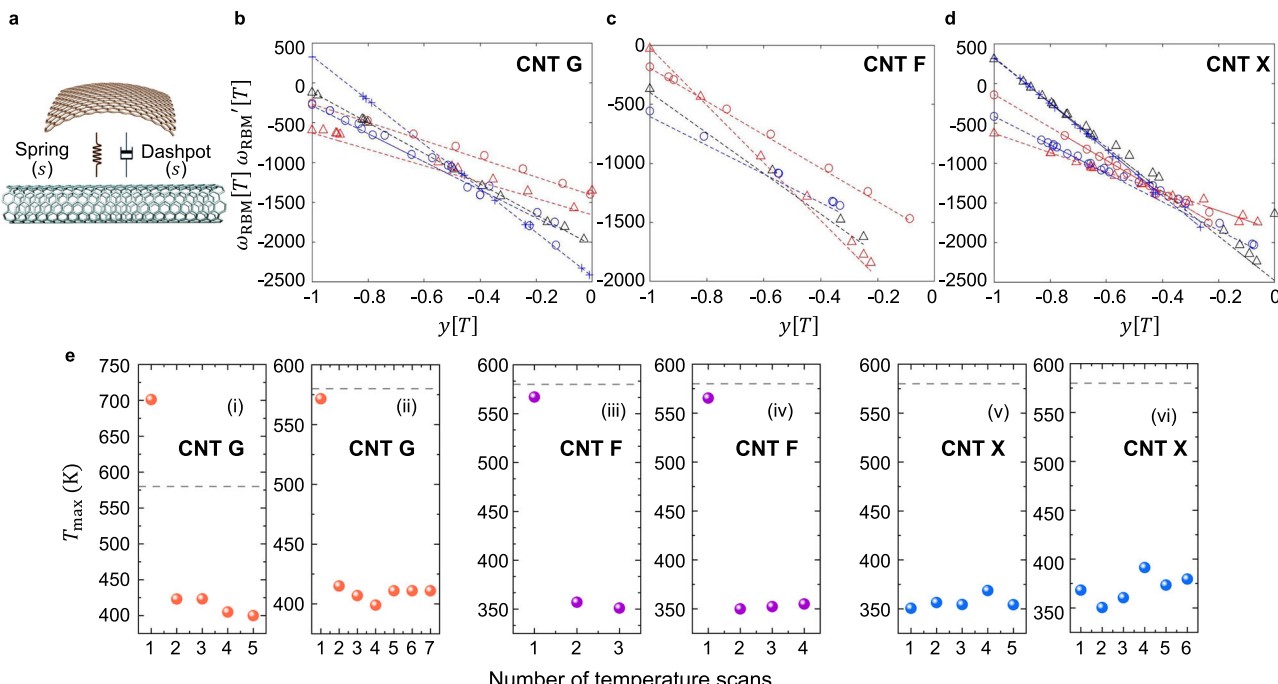

**Fig. 3 | Damping as the source of the large RBM shift with temperature. a** The radial force balance on the oscillating shell includes terms for the restoring force exerted by the graphitic ribbons (spring constant $\omega_0^2$) and damping function $b[T]^2$, which accounts for energy dissipation with oscillation (dashpot). $s$ denotes solid components from the graphitic ribbon. **b** The derivative test applied to CNT G and the 5 scans in Fig. 2a. The downward trend means that the second derivative is negative for all $T \leq T_{max}$, ruling out fluid adsorption, and justifying the linear expansion of $b[T]$. The linearity means the damping constant ($\Delta b^2$) can be expanded in temperature via Eq. (4) with constant $b'$ ($R^2 = 0.98, 0.90, 0.98, 0.97$, and $0.97$ for data symbols of red circle, black triangle, red triangle, blue circle, and blue cross, respectively) (Supplementary Table 12-3). **c** Same results for the 4 scans of CNT F in Fig. 2c with a systematic variation of the negative slope (equal to $b_{max}^2$) suggesting variation of this constant from scan to scan ($R^2 = 0.99, 0.99, 0.99$, and $0.99$ for data symbols of red circle, black triangle, red triangle, and blue circle, respectively) (Supplementary Table 12-2). **d** The 5 scans of CNT X in Fig. 2b also show negative linearity, demonstrating the same mechanism ($R^2 = 0.87, 0.95, 0.94, 0.98$, and $0.90$

for data symbols of red circle, black triangle, red triangle, blue circle, and blue cross, respectively) (Supplementary Table 12-1). **e** $T_{max}$ is seen most commonly at an initially high temperature between 550 K to 800 K upon the first heating scan, with subsequent scans then showing a consistent reduction to a reproducible value between 320 to 430 K. Scans (i) and (ii) correspond to CNT G at two different observation windows at $1.2 \times 10^{-6}$ bar showing the initially high $T_{max}$. The dotted line is the predicted $T_{max}$ of 580 K for a CNT under 22 pN tension and a CTE of $2 \times 10^5$ K$^{-1}$. The steady state $T_{max}$ corresponds to approximately a 50% strain reduction after initial heating. Scans (iii) and (iv) for CNT F at two different locations at $1.2 \times 10^{-6}$ bar show the same initially high $T_{max}$ followed by a reduction to a value consistent with a 75% reduction of initial strain. Scans (v) and (vi) for CNT X do not show the initially high $T_{max}$, however, an initial focus at high laser power may obscure it. The repeated values of $T_{max}$ fall into the low range observed for other CNTs and correspond to a 25% strain reduction from the initial self-tension estimate. Source data are provided as a Source Data File.2.

yet used quantitatively describe the RBM (Supplementary Text 6 and Supplementary Fig. 6−1). Here, $\bar{b}$ is the specific damping and $\rho$ is the axial CNT carbon density.

$$\left(\frac{\beta}{r^2} + \gamma\right)w[t] + \frac{\bar{b}}{2\rho}w'[t] + w''[t] = \omega_0^2 w[t] + 2bw'[t] + w''[t] = 0 \quad (2)$$

This Eq. (2) models the two shells of the DWNT as a single composite shell with a radial displacement $w[t]$ for simplicity. The more complex two-shell system is considered in Supplementary Text 7, where we show that it does not change the conclusions, with an error <2 cm$^{-1}$.

The solution to Eq. (2) yields the RBM frequency ($\omega_{RBM}$) as

$$\omega_{RBM}[T] = \sqrt{\omega_0^2 - b^2} \quad (3)$$

This basic force balance with $b = 0$ has been applied to describe the RBM shifts caused by a surrounding solvent[13,17,20,28,30], high pressure[18−20,31], and bundling[9,10,32]. From Eq. (2), a shift of the RBM due to environmental coupling can be caused by changes in the spring contribution ($\gamma$) or damping coefficient ($b$) (Supplementary Text 8: a and b).

The 9−14 cm$^{-1}$ RBM ($\omega_L$) change is larger than what has been measured between strongly coupled CNT bundles and the pristine vacuum state, indicating that the range would correspond to a completely disconnected spring if attributed to only $\gamma$. Additionally, a temperature dependent softening of the ribbon spring constant would have to decay with a non-physical power law dependence, as $T^n$ with $n > 1$ for a negative first and second derivative to be observed (Supplementary Text 8: c). In contrast, the magnitude and direction of the shift are consistent with previous estimates of the specific damping ($b$) for CNTs. The outer shell of CNT G (35,6) has a carbon atom mass of $7.2 \times 10^{-15}$ kg m$^{-1}$ and exhibits an estimated frequency damping of $(\omega_{RBM, max})^2 - (\omega_{RBM, min})^2 = 1{,}969$ cm$^{-2}$ or $44.4$ cm$^{-1}$. This predicts 9.5 mN s m$^{-2}$ as the specific damping change, in close agreement with the value of 8.3 mN s m$^{-2}$ calculated by Kumar et al.[9] for smaller diameter SWNT. These observations motivate a focus on a change in the damping term in Eq. (2) above as the source of the trajectories (Supplementary Text 9).

The temperature dependence of the damping can be directly obtained by the derivative of the square of Eq. (3). In terms of the scaling factor, $y[T]$ (Eq. (1)), and assuming that $\gamma$ remains approximately invariant with temperature:

$$\omega_{RBM}[T]\omega_{RBM}'[T] = -b[y[T]]b'[y[T]]y'[T] \quad (4)$$

We find that a plot of the right-hand side (RHS) versus $y[T]$ is consistently linear for all temperature scans, indicating that $b[T]$ is linear, and increasing towards $T_{max}$ (Supplementary Text 11). The slope of the RHS of Eq. (4) against $y[T]$ is negative when the coupling is attributed to the case of non-zero damping. This appears to be consistent for all trajectories studied in this work for CNT G, X, and F (Fig. 3b–d and Supplementary Text 2-4). Moreover, the linearity of the dependence motivates a linear expansion for $b[T]$, increasing with temperature towards $T_{max}$.

## The $T_{max}$ Cusp and minimum frequency

The unique features of this trajectory, including the downward concavity and cusp at $T_{max}$ seem to only be observed for CNTs suspended across the TEM window. For example, these features are not observed during the temperature cycling of CNTs bound to a substrate in past[24] or current experiments (Supplementary Text 1: a). Also, heating experiments on suspended CNTs in other studies have not produced any observable shift of the RBM[22], suggesting that the unique features of the particular substrate and TEM window employed in this work enable the observations. Like suspended graphene, CNTs stretched across such windows are under self-tension through vdW adhesion to the substrate window or through thermal contraction of the substrate from synthesis. Hence, this tension and the resulting strain can serve as potential sources of the damping change. Bunch et al. calculated a tension of 13 nN for a 2 μm wide graphene resonator[33]. Scaled to a 3 nm DWNT, the initial 33 pN tension ($Tn_o$) translates into an axial lattice strain of 0.61% at $T_o$ using a Young's modulus ($M$) of 0.5 GPa[34]. Subsequent heating of a CNT segment with the size of the Raman spot ($S = 2 \mu m$), as in this work, necessarily relaxes the tension through thermal expansion once the limit of vdW contact with the substrate is reached, until the strains equilibrate at a limiting temperature $T_{max}$ (Supplementary Text 10) via

$$T_{max} = \frac{Tn_o}{MS\alpha} + T_o \tag{5}$$

Deng et al. measured $\alpha$, the coefficient of thermal expansion (CTE) as $2 \times 10^{-5}$ K$^{-1}$ for a DWNT[35]. Using Eq. (5), this corresponds to a $T_{max}$ of 580 K, consistent with values observed in the initial heating cycle, as shown in Fig. 3e(ii)–(iv). If the initial heating cycle relaxes the $Tn_o$ tension value by 50% (to 11 pN), the resulting threshold temperature falls to 433 K, consistent with persistent $T_{max}$ values for CNT G in Fig. 3e(i)(ii) and CNT F in Fig. 3e(iii)(iv). CNT X does not exhibit the initially high $T_{max}$ in Fig. 3e(v)(vi), however, this self-tension may be relaxed due to an initial focus at high laser power (>0.3 mW). At this power, the heating of the CNT X is already substantial, impeding the subsequent observation of the initial trajectory as observed on CNT F and CNT G. In contrast, the initial power used for CNT F and CNT G is 0.01 mW. Equation (5) provides accurate estimates for the $T_{max}$ values observed in this work, and the mechanical mechanism is consistent with the regularity at which $T_{max}$ values can be repeatedly observed through heating or cooling cycles, as seen in Fig. 2a–c. Electron-beam cutting of the 1 mm CNT on both of its ends appears to eliminate all observations of $T_{max}$ suggesting a relaxation of tension of the CNT across all of the wells.

## Strain dependent lattice coupling

The RBM shifts from its highest value at $T_o$, corresponding to the weaker damped state $b_{min}$, to the lowest at $T_{max}$ corresponding to the strongest damping, $b_{max}$, after which, for $T > T_{max}$, the frequency remains invariant. From Eq. (1) (scaling factor, $y[T]$) and Eq. (3), expanding the damping constant linearly yields a hyperbolic expression for the RBM shift that can be used to fit the trajectories as follows,

$$\omega_{RBM}[T] = \sqrt{\omega_o^2 - (b_{max} + \Delta b y[T])^2} \tag{6}$$

This expression describes all 93 scans for CNT G, X, and F (red curves, Figs. 1e and 2a–c and Supplementary Text 2). Damping is a dynamic property related, in this case, to the material friction generated as the radial velocity of the vibrating CNT is damped by the jostling of the ribbon around it. Kitt et al.[36] found that lattice strain decreases friction at the graphene interface by reducing the available contact area, anticipating the reduced damping in this current system at the high strain state at $T_o$. As the strain relaxes on the approach to $T_{max}$, the thermally expanded ribbon can more completely adhere or even lattice register with the now more compliant CNT. In this way, the temperature can mediate the coupling of the graphitic ribbons between the weakly and strongly coupled states as the system passes from $T_o$ to $T_{max}$.

## The maximally damped limit

The RBM reaches its minimum frequency ($\omega_{RBM, min} = \sqrt{\omega_o^2 - b_{max}^2}$) at zero tension for all $T > T_{max}$, and the reproducibility of this limit is striking as different sections of the same CNT are scanned. We plotted the square of the maximum damping ($b_{max}^2$) versus the square of the net spring constant ($\omega_o^2$) for every scan of each DWNT in this work (Fig. 4a–c). The data disperse linearly, especially for CNT X and CNT F (Fig. 4b, c), suggesting:

$$b_{max}^2 = \nu \omega_o^2 + l \tag{7}$$

where $\nu$ is the dimensionless slope (cm$^{-2}$/cm$^{-2}$) with intercept $l$. As $b_{max}$ changes from scan to scan, there appears to be a compensatory change in $\omega_o$. This is seen in the trajectories as $b_{max}$ governs the concavity of the downward curvature. Along each line in Fig. 4a–c, stage pressures vary from 1 to $10^{-5}$ mbar, indicating no influence of pressure. The dimensionless slopes ($\nu$) are also similar at 1.006, 0.996, and 0.945 for CNT F, G, and X, respectively. This ratio is the differential change in maximum coupling $(\Delta b_{max})^2$ for a change in spring constant $(\Delta \omega_o)^2$. The invariance implied by the linearity in Fig. 4a–c suggests that as more graphitic ribbons attach to the segment in the probed spot, or as a given ribbon creates more tethers with the CNT surface, the increased damping suppressing the frequency is compensated by a net stronger restoring force. Each attachment brings both coupling types, resulting in an invariant minimum RBM frequency ($\omega_{RBM, min}$). In contrast, $\Delta b$ values exhibit the stochasticism expected from fluctuating attachments frustrated by the strained underlying lattice, contributing weaker damping at $T_o$, with a mean of 41.8 cm$^{-1}$ and standard deviation of 11.5 cm$^{-2}$ (Fig. 4d).

Quantitatively, the dispersion along the linear trajectory for $n$ graphitic attachments can be described by:

$$\omega_{RBM, min} = \sqrt{\left(\frac{\beta}{r^2} + n^2 \gamma_i\right) - n^2 b_{max,i}^2} = \sqrt{\omega_o^2 - b_{max}^2} \tag{8}$$

where $\gamma = n^2 \gamma_i$ and $b_{max} = n b_{max,i}$. A constant slope $\frac{d(b_{max})^2}{d(\omega_o)^2} = \frac{1}{\gamma_i}(b_{max,i})^2$ is predicted and hence the linearity in Fig. 4a–c is expected (Supplementary Text 10: a).

We thermally annealed CNT X for 12 h at 373 K and $3.2 \times 10^{-5}$ bar and, to our surprise, observed no noticeable change in the RBM trajectories or $T_{max}$. By contrast, we observe a change in the shape of the trajectory post-annealing upon the application of Eq. (3) to the data (Fig. 4e). The trajectory changes such that the $b_{max}$ and $\omega_o$ values correspond at a 16% decrease in the number of tethers upon annealing.

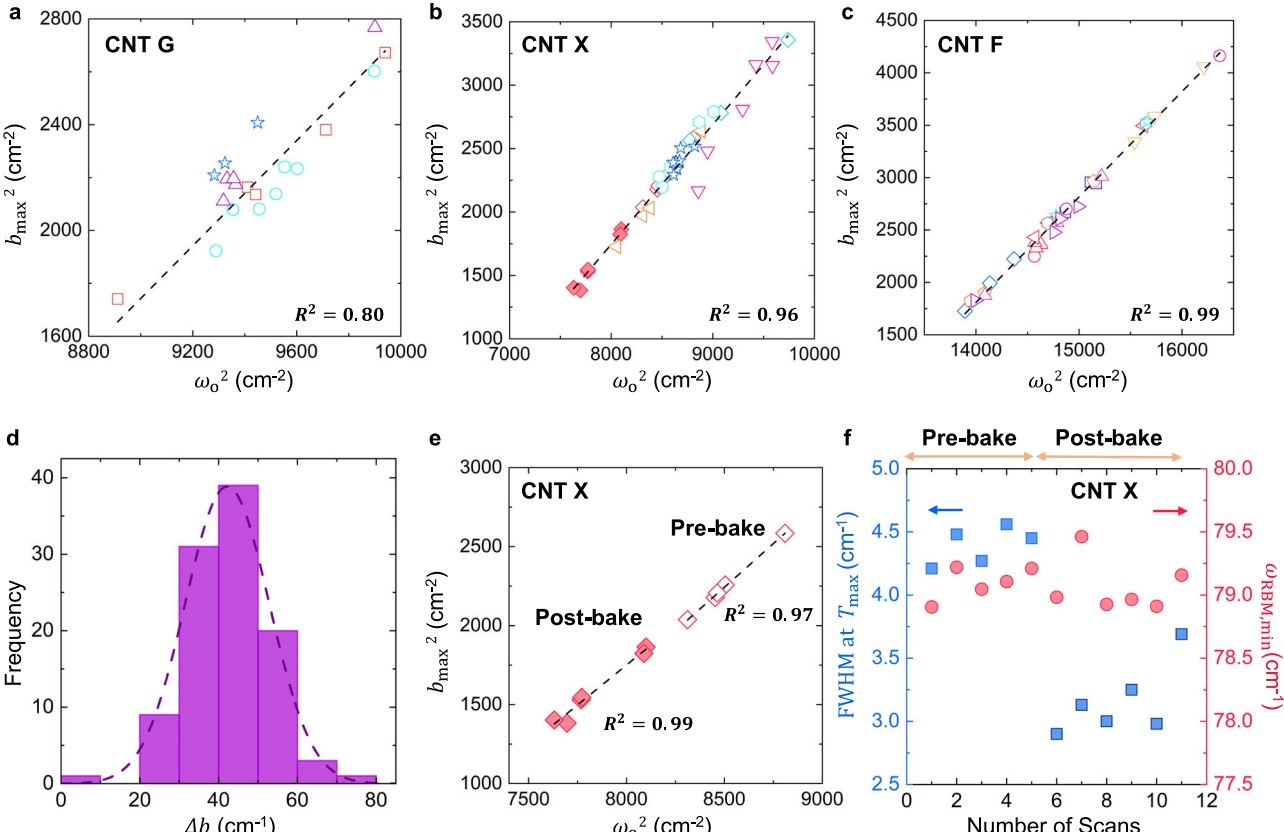

**Fig. 4 | Relationship between the maximum damping limit ($b_{max}^2$) and spring constants ($\omega_o^2$) for a series of DWNTs. a** The parameters (Supplementary Table 12-3) from the regression of the series of scans in Fig. 2a are seen to disperse approximately linearly ($R^2 = 0.80$) for the CNT G (29,1)@(35,6). **b** The linearity is more pronounced ($R^2 = 0.96$) for CNT X (28,5)@(28,11). Stage pressure spans $10^{-8}$ to $10^{-3}$ bar with no apparent trend (Supplementary Table 12-1), except a set (red filled diamonds) after 12 h of baking out at 100 °C and $3.2 \times 10^{-5}$ bar (Supplementary Table 12-4). **c** The linearity is also pronounced ($R^2 = 0.99$) for CNT F (19,3)@(22,11), again showing no apparent trend with stage pressure but with points from the same location clustering along the line represented as different data sets (Supplementary Table 12-2). The different datasets and symbols in Fig. 4a–c represent values from a series of repeated scans at different locations on the same CNT (Supplementary Table 12-1–12-4). **d** Values for $\Delta b$ ($b_{max} - b_{min}$) for all three DWNTs approximate a normal distribution (dash line) with a mean 41.8 (cm$^{-1}$) and standard deviation of 11.5 (cm$^{-1}$)$^2$. **e** Thermal annealing or baking out for 12 h at 100 °C and $3.2 \times 10^{-5}$ bar had the only demonstrable effect on the dispersion of these two mechanical parameters with a significant reduction in $b_{max}$ and $\omega_o$ after processing. The data before (Pre-bake, $R^2 = 0.99$) and after (Post-bake, $R^2 = 0.97$) still fall identically on the line for CNT X in **b**, suggesting only a reduction in the number of graphitic tethers. The different symbols in **a–d** represent Raman scans at different spots in the same CNT and vacuum conditions. **f** FWHM at $T_{max}$ and $\omega_{RBM,min}$ of CNT X change upon thermally annealing. Scans 1-5 are pre-bake (as-grown) CNT X. After 12-hr annealing of CNT X (scans 6-11), $\omega_{RBM,min}$ remains nearly invariant and FWHM becomes narrow. This result is consistent with a decrease in the inhomogeneity of the system and in the number of carbonaceous tethers (Supplementary Table 12-1 to 12-4 and Supplementary Fig. 12-2). Source data are provided as a Source Data File.

While the expectation is that different densities of graphitic ribbon impurities should be encountered at each location or from scan to scan, Eq. (8) predicts no change to the limiting frequency as observed in this work (Fig. 2). All three CNTs exhibit negative intercepts ($l$) of −12,282 cm$^{-2}$, −7220 cm$^{-2}$, and −5817 cm$^{-2}$ for CNT F, G and X, respectively. Equation (8) anticipates this as the uncoupled frequency limit, $\frac{\sqrt{\beta}}{r}$. Since all three have chirality-assigned shells, $\beta$ can be estimated from fundamental mechanical properties (Supplementary Text 11: b and Supplementary Table 11-1).

It is tempting to view the invariant region $T > T_{max}$ as a clean or uncoupled state at high temperature[13,28], but both theory and experiment assert that the opposite is true, if counter-intuitive. At $T_o$, the system is actually in a weaker damped state compared to $T > T_{max}$, where damping appears to saturate to a maximum value that is repeatedly observed. This discovery reverses the expectation that at high temperature, the invariant region is a pristine or uncoupled state. Instead, temperature appears to lock the system in a state of maximally damped frequency (Supplementary Text 12: a). The full width at half

maximum (FWHM) of the RBM peak, which is the difference between the two RBM frequencies at which the intensity equals half of its maximum value, is broadened at room temperature, when $b$ is inhomogeneous and narrow at high temperature, when a uniform, limiting $b_{max}$ is reached, consistent with an inhomogeneous broadening mechanism. Additionally, the FWHM varies considerably from spot to spot on the same CNT, while at the same time, the RBM remains largely invariant at $\omega_{RBM,min}$ (Supplementary Text 12 and Supplementary Fig. 12-1). We find, in fact, that the FWHM of the peak scales with the value of $b_{max}$ for that particular scan, which we note is also a measure of the density of carbonaceous impurities in that particular segment (equal to $nb_{max,i}$). The fact that the FWHM at high temperature does not converge to a minimal limiting value for a given CNT is evidence that it remains environmentally coupled to the entity causing the residual variation. Further, we note that the 12-h annealing experiment of CNT X resulted in a decrease of the limiting high-temperature FWHM from $4.40 \pm 0.15$ cm$^{-1}$ to $3.16 \pm 0.29$ cm$^{-1}$, consistent with the decrease in $n$, the number of carbonaceous tethers (Fig. 4f and Supplementary Figs. 12-2 and 12-3) and persistent environmental coupling.

## A new understanding of one-dimensional vibrational coupling

These results have significant implications for understanding the vibrational coupling of one-dimensional systems, particularly in the critical radial direction. The general picture of an increase in RBM frequency as associated with increased coupling from solvent or inter-tube interactions may be revised in light of these findings. As a means of semi-quantitatively measuring the amount of coupling mass to the CNT interior[12] or external surface[11], the RBM frequency should be described by the full dynamic force balance in the radial direction. With the inclusion of the damping term, the differential change in spring-to-damping constants brought by each coupling tether determines the observed frequency, not the net addition of mass.

This new understanding explains some basic puzzles observed throughout the literature and revises our understanding of pristine or uncoupled systems. For example, Liu et al.[37] examined the RBM frequencies of a series of ED-assigned DWNTs in vacuum, extracting shell-shell coupling constants (Supplementary Text 13). We obtain similar constants for CNT F, G, and X in this work using the strongly coupled limit ($T > T_{max}$) as the vibrational reference (Supplementary Fig. 13-1). In this way, past experiments assumed to be pristine can still allow for evidence of the same environmental coupling observed in this work. Kumar et al.[9] pulled apart bundled CNTs, finding an upshift, instead of an expected downshift, in RBM frequency in the extracted state. Their attribution to a damping term is now quantitatively described and validated in this work. As discussed above, the damping coefficient of CNT G is 9.5 mN s m$^{-2}$, and more details are provided in Supplementary Text 9. The deconstruction of a CNT bundle could allow carbonaceous residues to adsorb to the individual sidewalls, explaining the upshift.

The force balance in Eq. (2) can be expanded to any number $N$ of parallel coupling elements, where each is modeled as a generic Zener viscoelastic element, $j$, known to capture the dynamics of a wide range of materials[38].

$$\sum_{j=1}^{N}\left(\frac{\omega_{1,j}^2\omega_{2,j}^2}{\omega_{1,j}^2+\omega_{2,j}^2}\right)w[t] + \sum_{j=1}^{N}\left(\frac{\omega_{1,j}^2}{\omega_{1,j}^2+\omega_{2,j}^2}\right)4bw'[t] + w''[t] = 0 \qquad (9)$$

For the graphitic ribbons analyzed in this work, we find $\omega_1^2 = \omega_2^2 = 2\omega_0^2$ and the force balance simplifies to Eq. (5) in the absence of any other sources of environmental coupling. Equation (9) predicts that the observed RBM frequency will be dominated by the largest magnitude terms $j$ in the force balance, quantitatively explaining, for the first time, why there is no apparent mass sensitivity for an unopened CNT on a substrate even after complete immersion under water (Supplementary Text 1: a) despite the fact that more than 80% of the external area remains exposed for potential coupling to the fluid.

Equation (9) predicts that the addition of fluid to the interior of the CNT, with a dominant spring constant, should significantly alter the temperature trajectory. The derivation in Supplementary Text 6: a predicts an inversion of the second derivative from negative to positive. We used a focused-ion beam (FIB) to cut open arrays of CNTs otherwise suspended, as described above. The open arrays were exposed to water vapor to fill the CNTs, then sealed with TorrSeal™ vacuum sealant before measuring RBM thermal trajectories under vacuum ($10^{-6}$ bar) (Fig. 5a–e, Supplementary Fig. 14-1, Supplementary Text 14: a, and Supplementary Movie 2). We find that the resulting RBM trajectories are notably different with a concave up curvature in the high temperature limit, indicating a positive second derivative, and a smaller downshift (~6 cm$^{-1}$) compared to the mechanical effect described above (Fig. 5f). There is no characteristic $T_{max}$ cusp as predicted. We also find that these RBM trajectories can change in the magnitude of the RBM shift from scan-to-scan, as shown in Fig. 5f over 7 scans in order, converging to a consistent trend, unlike the mechanical effect reported above. Equation (9) above indicates that distinct mass couplings (graphitic ribbons plus interior water) do not shift additively; rather, the strongest coupling (interior water)

dominates the shift (Fig. 5f). Future works will describe this combination coupling effect quantitively.

These results are consistent with a fluid-filled CNT system that produces RBM thermal trajectories governed by internal fluid adsorption and desorption within the CNT confined space, which is distinct from the strain-dependent coupling mechanism observed in unopened and partially suspended CNTs elaborated above. The concave-up trajectories can be described by the Langmuir isobar adsorption model along with the RBM conversion to $q[T]$ detailed in Supplementary Text 14: b and Eq. (10). The experimental data exhibit a reasonable fit in the linearized form Eq. (11), and the negative slope correlates with the heat of adsorption ($\Delta H$) of water in the confined geometry (Fig. 5g and Supplementary Figs. 14-2 and 14-3). A similar positive second derivative trend is also observed in the system of the substrate-supported interior water-filling CNT (Supplementary Fig. 14-4).

$$q[T] = \frac{\omega_{RBM}^2 - \omega_V^2}{\omega_L^2 - \omega_V^2} = \frac{e^{\frac{\Delta H}{RT}}PK_o}{1 + e^{\frac{\Delta H}{RT}}PK_o} \qquad (10)$$

where $q[T]$ is the surface coverage, $\omega_V$ (at which $q = 0$) is the lowest observed RBM frequency, $\omega_L$ (at which $q = 1$) is the highest observed RBM frequency, $K_o$ is the equilibrium constant, $\Delta H$ is the heat of adsorption, and $P$ is the fluid pressure.

This allows the linearization of the data such that:

$$\ln\left(\frac{1}{q[T]} - 1\right) = -\frac{\Delta H}{R}\frac{1}{T} + \ln(PK_o) \qquad (11)$$

Lastly, the concave-up trajectories also match those measured for CNTs supported on a substrate, FIB cut opened, and measured under a layer of water at 1 atm (Supplementary Text 14).

The RBM thermal trajectories that result from interior filling with water, tracing the Langmuir isobar, help to elucidate our previous observations of phase changes of confined water in a series of 6 single and double-walled carbon nanotubes[24]. Because the CNTs were substrate-supported, the strain-dependent mechanism elucidated above is not operative. In prior work, we observed four CNTs with low-to-high density (vapor-liquid) transitions exceeding 50 °C, with two above 150 °C[24]. The RBM frequency measurements in this current work are more precise at 0.2 cm$^{-1}$ resolution, and the isobars measured in Fig. 5 exhibit a more nuanced picture of the fluid phase transition, which occurs over a broad window of temperatures. Instead of a sharp boundary, the fluid occupancy decays exponentially following a quantitative theory, the Langmuir isobar adsorption model (Eq. (10) and Eq. (11)). However, at 150 °C the data indicate that the higher density (liquid) water fraction that exists is still at 20% occupancy, reflecting the stability due to confinement originally postulated[24]. Future work will use the new techniques developed herein to map such phase transition isobars for CNTs as a function of diameter, allowing the resulting thermodynamic data to inform a confined fluid equation of state[39,40].

## Discussion

The series of ED-assigned DWNTs suspended across TEM windows under vacuum allowed for the careful study of environmental vibrational coupling with unprecedented precision. When such systems are opened and filled using saturated water vapor, the RBMs trace Langmuir isobars and exhibit elliptical trajectories, allowing measurement of the enthalpy of phase change. When otherwise isolated in vacuum and pre-tensioned across suspended TEM windows, the large 9 to 14 cm$^{-1}$ RBM downshift shift observed with increasing temperature is attributed primarily to a quantitative change in vibrational damping from adsorbed carbon. The analysis based on a harmonic oscillator model exhibits a hyperbolic trajectory that corresponds to a reversible

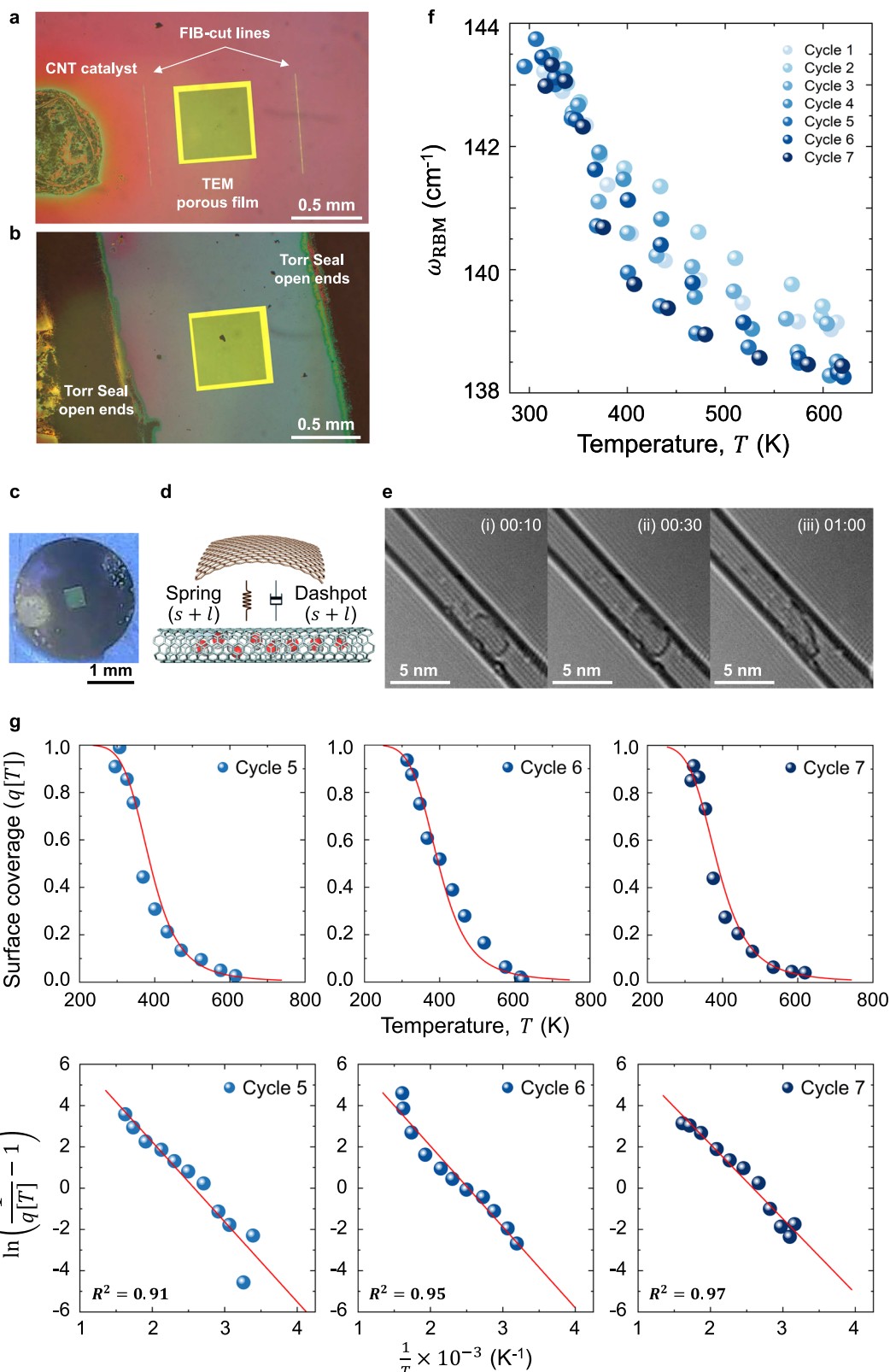

increase in damping, allowing for a quantitative description of experimental observations. Lattice strain from the initial self-tensioning of the suspended system is the likely source of the change in damping, as increasing temperature relaxes it to zero through thermal expansion, after which the frequency remains invariant. Overall, this mechanism and quantitative model significantly enhance our understanding of the environmental coupling of 1D

nanomechanical systems, providing the basis for emerging applications and the study of confined nanofluids.

## Methods

### Suspended CNT fabrication and imaging

Carbon nanotubes (CNTs) were grown on lithographically patterned silicon substrates using gas-flow aligned[23] chemical vapor

**Fig. 5 | Interior water-filled CNT sample fabrication and fluid isobar analysis. a** A microscopic image of FIB-cut opened CNT samples. Left: a drop-casted CNT catalyst; Squared window: 0.5 mm × 0.5 mm TEM window with 200 nm pores on a $Si_3N_4$ membrane; Both ends: FIB-cut lines. **b** A microscopic image of a Torr-sealed CNT sample. Following the CNT sample exposure to water vapor, Torr seal glues were applied outside of the TEM window to seal both ends of CNTs. **c** A picture of the interior water vapor-filled CNT sample. **d** A schematic illustration of a water-filled CNT system. *s* denotes solid components from the graphitic ribbons, and *l* represents liquid components from the internal fluid, depicted as red water molecules inside the CNT. Both components bring spring and dashpot contributing to the RBM shift, as described by Eq. (9). **e** TEM images of a dynamically formed entity from the electron beam within the interior CNT after a water-vapor filling procedure, assigned to a water-carbon oxidation product. TEM image at the time of (i) 10 s, (ii) 30 s, (iii) 1 min (Supplementary Movie 2). Not observed in an as-grown (empty) CNT. **f** The RBM of the interior wall displays reversible thermal trajectories and positive second derivative over 7 temperature cycles. **g** The experimental data (blue circles) were fitted to the Langmuir isobar adsorption as shown in the red curves and exhibited a reasonable fit in the linearized form indicated in the red lines (Supplementary Text 14). Source data are provided as a Source Data File.

deposition (CVD) with methane as a feedstock and iron as the catalyst, similar to our previous work[12,24,25,41]. Where indicated, CNTs were also grown suspended over holes in commercially available transmission electron microscopy (TEM) grids of variable pore size for comparison (PELCO®, Ted Pella, Inc). A solution of 25 Series APT carbon nanotubes (Nano-C) containing iron particles, Fe (<1 nm) was deposited on one end of the substrate by drop-casting or alternatively evaporated Fe was used as a catalyst. The specimens were annealed in air at 550 °C for an hour in an attempt to burn out amorphous carbon and impurities before the nucleation and growth of CNTs. The inclusion or exclusion of this annealing step did not appear to change the RBM trajectories studied in this work. CVD growth of the CNTs was conducted at 970 °C and $H_2$:$CH_4$ ratios of either 4 sccm:2 sccm or 8 sccm:4 sccm; there was no significant difference in growth results between the two ratios[12,24,25,42]. The CVD growth process resulted in the formation of a sparse array of parallel, millimeter-long CNTs oriented in the direction of gas flow.

For the water vapor filling, the CNTs were subsequently cut into segments using focused ion beam (FIB) milling with the 30 kV $Ga^+$ beam of a dual-beam FIB/scanning electron microscope (SEM) instrument (FEI Helios Nanolab 600)[25] on the substrate portion of the array. The FIB-cut CNTs were exposed to the water vapor by placing them in a 100 % humidity chamber for 10 min and subsequently sealed at both open ends with a Torr-Seal glue (Agilent Technologies). Ultra-pure (ASTM Type II) water was used for water immersion. Samples were examined by S/TEM (Titan Themis Z G3 Cs-Corrected S/TEM) in the TEM mode at MIT Nano for TEM imaging as described in Supplementary Text 14:a.

### Raman spectroscopy

CNTs were located based on their characteristic Raman scattering response using a confocal micro-Raman spectrometer in back-scattering geometry equipped with 532 nm, 633 nm, and 785 nm laser lines (Horiba LabRAM HR Evolution with hole 500 μm, slit 150 μm, Olympus MPLFLN 50X air objective and Olympus LUMPLFLN 60× water-immersion objective, NA = 1.0)[26]. A motorized stage with 0.1 μm precision (Märzhäuser Wetzlar SCAN) was used to move samples in x, y, and z relative to the fixed laser beam. The temperature dependence of the G band for each CNT segment was calibrated by measuring Raman spectra as a function of substrate temperature $T_{stage}$, which was controlled using a temperature stage (THS350EV, Linkam Scientific). To minimize laser-induced heating, the sample was placed in a 1 atm gas environment (air, $N_2$, or Ar), and very low laser power (on the order of 10 μW) was applied to record these calibration Raman spectra. The frequency of the most intense G band component $\omega_G$ was determined via a Lorentzian curve fit of each spectrum, and the resulting $\omega_G(T_{stage})$ calibration data were fit to a line (Supplementary Text 10)[12]. In general agreement with the literature and previous reports, we typically found its slope $d\omega_G/dT \approx -0.02 \sim -0.03\,cm^{-1}/K$ but with variable intercept. To study the temperature-dependent radial breathing mode (RBM) trajectories in this work, isolated and suspended double-walled carbon nanotubes (DWNTs) were placed in vacuum up to $10^{-8}$ bar for all measurements, except where noted. In all cases, the local temperature of the CNT was varied by attenuating the incident excitation laser beam using a set of discrete and continuous optical density filters. We determine the local temperature of a CNT segment under study based on the measured G band frequency $\omega_G$ and its calibrated $d\omega_G/dT$ dependence in a method employed previously[12,24,25].

### Reporting summary

Further information on research design is available in the Nature Portfolio Reporting Summary linked to this article.

## Data availability

Source data are provided as a Source Data file. The data supporting the findings of this study are also available from the corresponding author upon request. Source data are provided with this paper.

## Code availability

The code supporting the findings of this study is available from the corresponding author upon request.

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

## Acknowledgements

This work was supported as part of the Center for Enhanced Nano-fluidic Transport (CENT), an Energy Frontier Research Center funded by the U.S. Department of Energy, Office of Science, Basic Energy Sciences under Award # DE-SC0019112. M.K. acknowledges support from the German Research Foundation (DFG) Research Fellowship KU 3952/1-1. This work was carried out in part through the use of MIT.nano's facilities. We thank Philippe Lambin for the Fortran code DIFFRACT. This work was partially supported by the Center for Nanophase Materials Sciences (CNMS), which is a DOE Office of Science User Facility.

## Author contributions

Y.-M.T., M.K., and M.S.S. conceived and designed the research. Y.-M.T. and M.K. synthesized the samples and constructed the platform with collaborative input from C.L.R., S.F., S.Y., and J.Chakraverty. Y.-M.T., M.K., and C.L.R. characterized CNT samples, including Raman spectro-scopy and electron diffraction (ED) assignments with contributions from J.F.Y.; Y.-M.T., H.L., X.X., A.P., K.S., A.M., J.A.H., and J.Cumings per-formed transmission electron spectroscopy (TEM) imaging, analysis, and insight into nanomaterials in vacuum systems. R.P.M., H.O., N.R.A., and D.B. provided expertize in confined fluid thermodynamics. Y.-M.T. and M.S.S. analyzed and interpreted the Raman trajectory datasets with insights provided by R.P.M. and M.K.; Y.-M.T. and M.S.S. wrote the manuscript. All authors reviewed and edited the manuscript, and agree with the analysis and conclusions.

## Competing interests

The authors declare no competing interests.

## Additional information

**Peer review information** : *Nature Communications* thanks Filippo Boi, Richard Martel and the other, anonymous, reviewer(s) for their contribution to the peer review of this work. A peer review file is available.

¹Department of Chemical Engineering, Massachusetts Institute of Technology, Cambridge, MA, USA. ²Department of Physics, Brown University, Providence, RI, USA. ³Department of Mechanical Science and Engineering, University of Illinois Urbana-Champaign, Urbana, IL, USA. ⁴Department of Mechanical Engineering, Stanford University, Stanford, CA, USA. ⁵MIT.nano, Massachusetts Institute of Technology, Cambridge, MA, USA. ⁶Department of Physics, University of Maryland, College Park, MD, USA. ⁷Department of Materials Science and Engineering, University of Maryland, College Park, MD, USA. ⁸Stanford Precourt Institute for Energy, Stanford, CA, USA. ⁹Department of Mechanical Engineering, Oden Institute for Computational Engineering and Sciences, University of Texas at Austin, Austin, TX, USA. ¹⁰Center for Nanophase Materials Sciences, Oak Ridge National Laboratory, Oak Ridge, TN, USA. ¹¹These authors contributed equally: Yu-Ming Tu, Matthias Kuehne. ✉e-mail: strano@mit.edu

