## [Peer Review File · Nature Communications]

Environmental Damping and Vibrational Coupling of Confined Fluids within Isolated Carbon NanotubesREVIEWER COMMENTS

Reviewer #1 (Remarks to the Author):

The manuscript reports on the temperature dependence of Radial Breathing Mode (RBM) frequencies in free-standing double-walled carbon nanotubes (DWNTs) analysed by employing transmission electron microscopy in vacuum. The authors show large hyperbolic downshifts of 10 to 15%, with the environmental coupling occurring to ribbon-like carbonaceous materials in the first case. One of the very interesting aspects of the work is that the graphitic ribbons have the potential to lattice register with the underlying CNT and influence the RBM frequency. Can the authors comment on which type of lattice configuration occurs between the CNT and the ribbon? Would this then result into a stacking-fault-like configuration? This is an aspect that could be described more in the manuscript. Comparative investigations are also reported on water-filled CNTs. A point that would be important to clarify in the manuscript is the presence of possible water intercalation within the CNT layers. Is the fluid confined only within the inner capillary? Are there ribbon-like components also in the water-filled CNTs? The structural comparison of the two systems needs to be improved for clarity. Overall, the work is of significance and provides new insights on the properties of carbon nanotubes and hybrid carbon nanotube/ribbon systems. I recommend publication after the above mentioned revisions.

Reviewer #2 (Remarks to the Author):

In the manuscript "Mapping phonon hydrodynamic strength in micrometer-scale graphite structures" the authors investigate multiple sets of suspended double wall nanotubes (DWNTs), under vacuum conditions, explored via raman spectroscopy. For the case of closed, isolated, pre-tensioned DWNTs, they report a Radial Breathing Mode (RBM) downshift ($9-14 \text{ cm}^{-1}$) upon increasing the temperature of the DWNT. The authors rationalize the experimental findings on the basis of a simple damped oscillator model (spring-dashpot model). They attribute the downshifts to the damping to the adsorbed carbon present on the SWNTs. The authors suggest the damping changes with temperature because of the strain relaxation with increasing temperature up to a critical temperature beyond which the strain is completely relaxed. The authors then investigate the case of open DWNT filled with saturated water vapor. They assume the adsorbed fluid modifies the spring constant in a way proportional to the surface coverage fraction. Further assuming the latter to follow Languimir isobars, they are able to rationalize the RBM downshift VS temperature curves for this case. At the end of the manuscript the authors revisit experimental results from the literature, and discrepancies among them, at the light of their model.

The results are new and provide a new sight on a relevant topic, that of the acoustic damping/coupling of CNT mechanical vibrations, for which contradicting results/interpretations are provides in the literature.

The manuscript message is timely and of impact for the applied physics/nano-science community in general, its content being of interest to nano-acoustics, nano-device physic and nano-sensor technology at large.

The adopted methodology is sound. I particularly appreciated the rationalization based on a simple damped oscillator model. This simple model is valuable for experimentalist and theoretical physicist alike. This is an add-on with respect to a relevant portion of the recent literature where the trend is to rely in unnecessary complicated simulations, making it hard to unveil the underlying physics.

The authors provide a plethora of measurements and do not, so as to speak, "hide anything under the carpet". They are very frank and thorough in pointing outliers and aspects that might not fit within their general explanation. The authors provide a copious number of details, certainly enough for the work to be reproduced.

The bibliography is pertinent and up to date.

That said there is one major and some minor ones, which, in my opinion, the authors should address.

Major issue: readability.

The authors try to keep the explanation colloquial in the main text, referring the reader to Supplementary Information (amounting to 63 pages organized in 12 sections). In some portions the main text loses self-consistency, equations being discussed without the quantities being defined (if not in SI). For instance, just for the sake of exemplification, on page 7, lines 142-146 are incomprehensible. The scaling concept and the quantities therein addressed fall out of the blue. The good-willed reader has to work his way through jumping back and forth from the main to SI to grasp the meaning and this happens multiple times throughout the text (for instance when introducing the Languimir isobar adsorption model on page 1 line 381, the connection to the oscillation frequency is obscure without going through SI).

The reader has to plunge into the task of merging the infos from the Main text and SI, the latter containing key explanations for the comprehension. I did go through the entire process and, at the end, everything seems consistent, but it was an exhausting stop and go reading process, the explanations being too fragmented.

Minor issues

1. When discussing the lack of the initial T_{\max} for CNT X (page 10 line 216) the authors suggest this may be due to relaxed tension due to an initial focus at high laser power. Could the authors be more specific? For instance (a) providing the expected temperature increase and (b) why the focused beam power differs from the one adopted on the other samples?

2. In my opinion, when discussing previous literature results at the light of the new model, the discussion is somewhat too qualitative.

For instance, when discussing previous data from Kumar et al. (page 16, line 248-352) could the authors provide actual numbers to substantiate the statement: "Their attribution to a damping term is now quantitatively described and validated in this work."

Alike, when discussing the contribution of multiple springs (Eq. 9), the authors state: "The smaller shift is consistent with some combination of a larger spring constant 373 and decreased damping from the fluid compared to the graphitic ribbon". Could the authors elaborate further providing some figures?

3. Poor figure readability. Figures are generally too small to be readable. This is especially true for: Fig. 1 panel (b), (d) and f (vi); Fig. 2 first column (diffraction patterns)

4. Eq. S5-1 is inconsistent with Eq S5-8 (the square root) and, both, seem inconsistent with S6-2 (β is defined as the frequency on page 23 not frequency square).

More of a scientific curiosity for future studies. An alternative technique to address vibration damping in nanostructures is time-resolved optical spectroscopy. With this technique the breathing mode is directly excited by a pump beam and the resulting trace allows to directly visualize the displacement oscillation and damping in the time-domain. This technique has been applied to investigate individual nanostructures too. Are the authors aware of similar measurement carried out on single suspended CNT to retrieve the RBM frequency and damping vs temperature? If yes, it might be worth mentioning the reference to trigger the interest of the ultrafast community.

Summarizing, for the above-mentioned reasons I recommend Major revisions.

Reviewer #3 (Remarks to the Author):

The paper by Tu et al. presents Raman experiments on free-standing individual carbon nanotubes (CNTs) assigned by electron diffraction. Reversible RBM shifts with temperature are measured in vacuum and assigned to interactions with the environment (adsorbed carbon species). The first study presents 3 different DWCNTs first (Fig. 1-4) with characteristic red (down) shifting RBM with temperature in vacuum (10^{-3} - 10^{-8} bar). The RBM shifts have hyperbolic shapes up to a max temperature of ~ 350 - 400 K, after which the RBMs remain constant. This behavior is ascribed to vibrational damping from graphitic adsorbates located on the CNT external walls. Using damped oscillator model, the authors argue that the RBM shifts are due to the dynamic force balance in the radial direction and not from additional mass onto the oscillator. While I appreciate the efforts behind these experiments and phenomenological model derivations, I am not sure the results represent a significant advance to the topic of nanotube sensing of confined fluids. RBM shifts by adsorbates such as water adsorption onto the inner or outer walls have been extensively studied in the past and analysis based on damping harmonic oscillator has also been done (e.g. in Ref. 9). The shifting trend described here from carbon contaminants at the surface of DWCNTs is probably original and the model makes sense, but the significance of this work is unclear. I do not recommend publication in Nature Communication. This work is of interest for a more specialized journal, but there are major issues that will need to be addressed, as detailed below.

1) Many Raman spectra involving different nanotubes and different temperatures were acquired. Except for Figure 1d, these Raman spectra are not shown, not even in the SI file. The only sequence in Figure 1d is very small and of low digital quality, which is problematic. Original spectra and fits to the experimental results should be presented more clearly.

2) Based on Figure 1d, RBMs are sharp at low T and more difficult to see at higher T, which makes me wonder how accurate the fits are to extract the positions and FWHMs for each run. Furthermore, the loss of intensity could also be due to a general loss of resonance, and this must be discussed somehow.

3) Considering that the inner wall of a DWCNT is closely coupled to the outer wall, an absence of discussion of the inner wall RBM is surprising. Moreover, it is unclear why there is a need to ED assign the DWCNTs here. This information is not used anywhere in the analysis provided (qualitative comparisons with literature values are made in the SI file but not used).

4) The base pressure for those experiments is rather poor, in mbar range, and far from UHV. Hence, I am not sure the authors can safely rule out gas adsorption at the nanotube surface in those conditions. Experiments on clean DWCNTs (exempt of carbonaceous contaminants at the surface) are also missing. This is needed to support the conclusion of a damping effect by graphitic ribbons.

5) Equation 5 to evaluate T_{max} deals with length units that can be problematic. The dimensions of the nanotubes are rather small, and this may give speculative estimates of T_{max} . Please explain more clearly the choices of length scales that were used to reach the 580K value.

6) The discussion on maximum damped limits (p. 12-14) and FWHM behavior (p. 15 and Fig. 4f) is very confusing and hard to follow. Quantitative results from actual spectra are not shown and it is hard to understand how inhomogeneity along the nanotubes can contribute to changes in FWHM or damping forces. The discussion related to the homogeneity and broadening with the number of carbonaceous tethers at T_o vs. T_{max} is unclear in the current form of the paper. Further experimental evidence is probably needed to support these claims.

7) RBM shifts with temperature in Figure 5 due to water confinement inside nanotubes is an extension of past work in Ref. 24. It is not clear what CNTs are studied here, and again no Raman spectra are presented. Moreover, I am not sure I understand the relation with the other experiments in Figure 1-4. What new knowledge was gain here compared to past studies on water is unclear. Is this the topic of another paper or simply an example of application of the damped oscillator model?

Reviewer #4 (Remarks to the Author):

The manuscript presents a comprehensive investigation into the Radial Breathing Mode (RBM) frequencies exhibited by isolated suspended double-walled carbon nanotubes when coupled with external environments and confined fluid within their inner cavities. The study utilizes transmission electron microscopy, electron diffraction, and micro-Raman spectroscopy with three laser excitations to analyze a series of nanotubes for statistical purposes. Additionally, supplementary information detailing the Raman setup and the precise protocol for recording the RBM frequencies is provided. A proposed model, integrating a harmonic oscillator and damping system, effectively explains the observed phenomena. Moreover, the paper is expected to appeal to a wider audience within the community interested in investigating filled carbon nanotubes. Notably, the mechanical properties of nanotubes may be influenced not only by solid species adsorbed on their outer surface but also by those encapsulated within their interior (molecules, carbon allotropes, crystals), potentially resulting in significant modifications in the RBM frequencies, including their cancellation. Overall, this is a very good manuscript.

With some enhancements, I recommend publication of this manuscript.

The abstract could be less technical or simplified for broader audience.

Regarding line 61, perhaps rephrase to avoid using 'absent' as follows: 'In the absence of an underlying substrate...'

Additionally, in line 167, Equation 2 is referenced without prior presentation.

In the caption of Figure 4, it would be beneficial to define FWHM (Full Width at Half Maximum) and specify what it pertains to.

Finally, could this model be applicable to single-walled carbon nanotubes and their environmental interactions?

RESPONSE TO REVIEWERS' COMMENTS

Reviewer #1 (Remarks to the Author):

Comment 1-1: The manuscript reports on the temperature dependence of Radial Breathing Mode (RBM) frequencies in free-standing double-walled carbon nanotubes (DWNTs) analysed by employing transmission electron microscopy in vacuum. The authors show large hyperbolic downshifts of 10 to 15%, with the environmental coupling occurring to ribbon-like carbonaceous materials in the first case. One of the very interesting aspects of the work is that the graphitic ribbons have the potential to lattice register with the underlying CNT and influence the RBM frequency.

Can the authors comment on which type of lattice configuration occurs between the CNT and the ribbon?

Would this then result into a stacking-fault-like configuration?

This is an aspect that could be described more in the manuscript.

Response 1-1: We appreciate the reviewer's insights and suggestions. We concur with the reviewer's comment regarding the stacking-fault-like configuration. The mismatched lattice strain arising from the initial self-tensioning of the suspended CNT is likely a stacking fault bounded by partial dislocations of carbon atoms between graphitic ribbons and CNTs at room temperature^{1,2}. As temperature rises due to local laser heating, the self-tension of CNT diminishes to zero at T_{\max} through thermal expansion. This heating process supplies energy that fosters the adhesion of graphitic ribbons onto the CNT surface, leading to an increase in the damping coefficient until reaching the characteristic temperature T_{\max} with invariant frequency. Kitt *et al.*³ demonstrate that lattice strain diminishes friction at the graphene interface by decreasing the available contact area, suggesting a reduction in damping in our system at the high strain state at T_0 . As the strain relaxes towards T_{\max} , the thermally expanded ribbon can more effectively adhere or even lattice register with the now more compliant CNT. Consequently, the temperature can mediate the coupling of the graphitic ribbons from weakly to strongly coupled states as the system passes from T_0 to T_{\max} .

We have revised the statement to incorporate the stacking-fault-like configuration.

“Damping is a dynamic property related, in this case, to the material friction generated as the radial velocity of the vibrating CNT is damped by the jostling of the ribbon around it. The mismatched lattice strain arising from the initial self-tensioning of the suspended CNT is likely a stacking fault bounded by partial dislocations of carbon atoms between graphitic ribbons and CNTs at room temperature^{1,2}. As temperature rises due to local laser heating, the self-tension of CNT diminishes to zero at T_{\max} through thermal expansion. This heating process supplies energy that fosters the adhesion of graphitic ribbons onto the CNT surface, leading to an increase in the damping

coefficient until reaching the characteristic temperature T_{\max} with invariant frequency. Kitt *et al.*³ found that lattice strain decreases friction at the graphene interface by reducing the available contact area, anticipating the reduced damping in this current system at the high strain state at T_o . As the strain relaxes on the approach to T_{\max} , the thermally expanded ribbon can more completely adhere or even lattice register with the now more compliant CNT. In this way, the temperature can mediate the coupling of the graphitic ribbons between the weakly and strongly coupled states as the system passes from T_o to T_{\max} .”

Comment 1-2: Comparative investigations are also reported on water-filled CNTs. A point that would be important to clarify in the manuscript is the presence of possible water intercalation within the CNT layers.

Is the fluid confined only within the inner capillary?

Response 1-2: Yes, we assign the addition of fluid as confined within the inner capillary of CNTs. This is based on the fact that water-filled suspended CNTs of this current work exhibit an O-H vibrational mode detected using Electron Energy Loss Spectroscopy (EELS) characteristic of confined water. These measurements are a part of a related but separate study, now under review in the Nature Journal and *arXiv paper*. We have not encountered any evidence for inter-collated water between the layers of the DWNT.

Comment 1-3: Are there ribbon-like components also in the water-filled CNTs?
The structural comparison of the two systems needs to be improved for clarity.

Response 1-3: Yes, the graphitic ribbons are also present on the exterior surface of a water-filled suspended CNT, as shown in the TEM image (Fig. R1, a large region of Fig. 5d and Video S2). As the bake-out experiment reported in Fig. 4e, these carbon species cannot be completely removed. We include the schematic illustration of water-filled CNT within Fig. 4e in the revised manuscript. Some smaller carbon fragments are occasionally visible in TEM in the CNT interior. The SI contains representative TEM images of both types of CNTs (filled and unfilled) explored in this work.

We have also clarified the description in this section.

“Eq. 9 predicts that the addition of fluid to the *interior* of the CNT, with a dominant spring constant, should significantly alter the temperature trajectory...as well as Eq. 9 above indicates that distinct mass couplings (graphitic ribbons plus interior water) do not shift additively, rather the strongest coupling (interior water) dominates the shift.”

Fig. R1| A large region of TEM image (Fig. 5d and Video S2) of a dynamically formed entity from the electron beam within the interior CNT. The graphitic ribbons exist on the surface of a water-filled suspended CNT.

Fig. R2| Schematic illustration of (a) graphitic ribbon-coupling CNT (Fig. 3a) and (b) the water-filled CNT (within Fig. 4e) systems studied in this work. s denotes solid components from the graphitic ribbon, and l represents liquid components from the internal fluid. Both components bring spring and dashpot contributing to the RBM shift, as described by Eq. 9.

Overall Comment: Overall, the work is of significance and provides new insights on the properties of carbon nanotubes and hybrid carbon nanotube/ribbon systems. I recommend publication after the above mentioned revisions.

Response: We greatly appreciate the reviewer's remarks on the importance and insights of the work. We thank and acknowledge the recognition of our two systems and mechanisms: the spring and dashpot model and the Langmuir confined fluid isobar model, presented in this manuscript.

Reviewer #2 (Remarks to the Author):

Overall Comment: In the manuscript “Mapping phonon hydrodynamic strength in micrometer-scale graphite structures, *Author correction: Anomalous Environmental Damping in Vibrationally Coupled and Water Filled Carbon Nanotubes*” the authors investigate multiple sets of suspended double wall nanotubes (DWNTs) under vacuum conditions, explored via Raman spectroscopy. For the case of closed, isolated, pre-tensioned DWNTs, they report a Radial Breathing Mode (RBM) downshift (9-14 cm^{-1}) upon increasing the temperature of the DWNT. The authors rationalize the experimental findings on the basis of a simple damped oscillator model (spring-dashpot model). They attribute the downshifts to the damping to the adsorbed carbon present on the SWNTs. The authors suggest the damping changes with temperature because of the strain relaxation with increasing temperature up to a critical temperature beyond which the strain is completely relaxed.

The authors then investigate the case of open DWNT filled with saturated water vapor. They assume the adsorbed fluid modifies the spring constant in a way proportional to the surface coverage fraction. Further assuming the latter to follow Languimir isobars, they are able to rationalize the RBM downshift VS temperature curves for this case. At the end of the manuscript the authors revisit experimental results from the literature, and discrepancies among them, at the light of their model.

The results are new and provide a new sight on a relevant topic, that of the acoustic damping/coupling of CNT mechanical vibrations, for which contradicting results/interpretations are provides in the literature.

The manuscript message is timely and of impact for the applied physics/nanoscience community in general, its content being of interest to nano-acoustics, nano-device physics, and nano-sensor technology at large.

The adopted methodology is sound. I particularly appreciated the rationalization based on a simple damped oscillator model. This simple model is valuable for experimentalist and theoretical physicist alike. This is an add-on with respect to a relevant portion of the recent literature where the trend is to rely in unnecessary complicated simulations, making it hard to unveil the underlying physics.

The authors provide a plethora of measurements and do not, so as to speak, “hide anything under the carpet”. They are very frank and thorough in pointing outliers and aspects that might not fit within their general explanation. The authors provide a copious number of details, certainly enough for the work to be reproduced.

The bibliography is pertinent and up to date.

That said there is one major and some minor ones, which, in my opinion, the authors should address.

Response: We greatly appreciate the reviewer’s positive comments and are enthusiastic about the reviewer recognizing the simple damped oscillator model, mechanisms, and broad prospects of our research in several fields, including applied physics and nanoscience. We also thank the

reviewer for emphasizing the methodology, environmental coupling mechanism, and confined fluid isobar model of our work. Specific comments and the improvement of manuscript readability and scientific contents are addressed below.

Major issue: readability.

Comment 2-1 (a): The authors try to keep the explanation colloquial in the main text, referring the reader to **Supplementary Information** (amounting to 63 pages organized in 12 sections). In some portions, the main text loses self-consistency, with equations being discussed without the quantities being defined (if not in SI). For instance, just for the sake of exemplification, on page 7, lines 142-146 are incomprehensible. The scaling concept and the quantities therein addressed fall out of the blue.

Response 2-1 (a): This is an outstanding suggestion by the reviewer. We appreciate the request for clarification and readability regarding the organization between the main text and **Supplementary Information (SI)**.

We have clarified the description and explanation of the scaling factor $y[T]$ as below,

“

$$y[T] = \frac{T - T_o}{T_{max} - T_o} - 1 \quad \text{Eq. 1}$$

The scaling factor $y[T]$ in **Eq. 1** is applied according to the experimental observation in the RBM trajectory of an unopened and free-standing CNT system. The RBM trajectories consist of a concave down curve towards a characteristic cusp at a limiting temperature, T_{max} , followed by an invariant RBM frequency for all $T > T_{max}$. Hence, this observation suggests the scaling factor as **Eq. 1**, and the $y[T]$ scales the transition of the damping term, b , with the temperature, T , according to the expression: $b = b_{max} + y[T]\Delta b$, where $\Delta b = b_{max} - b_{min}$ as well as b_{max} and b_{min} correspond to the damping at T_{max} and T_o , respectively. The features of these RBM trajectories of an unopened and free-standing CNT system are consistently observed in all 93 scans analyzed in this work (**Supplementary Text S2-S4**).”

We have also modified the sentence below to indicate the use of this scaling factor $y[T]$ for the RBM trajectory fitting.

“From **Eq. 1** (scaling factor, $y [T]$) and **Eq. 3**, expanding the damping constant linearly yields a hyperbolic expression for the RBM shift that can be used to fit the trajectories as follows,

$$\omega_{RBM}[T] = \sqrt{\omega_o^2 - (b_{max} + \Delta b y[T])^2} \quad \text{Eq. 6}$$

Comment 2-1 (b): The good-willed reader has to work his way through jumping back and forth from the main to SI to grasp the meaning and this happens multiple times throughout the text (for instance when introducing the Langmuir isobar adsorption model on page 1 line 381, the connection to the oscillation frequency is obscure without going through SI).

The reader has to plunge into the task of merging the infos from the Main text and SI, the latter containing key explanations for the comprehension. I did go through the entire process and, at the end, everything seems consistent, but it was an exhausting stop and go reading process, the explanations being too fragmented.

Response 2-1 (b):

We particularly thank you for your extra effort in finding and examining the essential explanations and details in the SI. We have revised a few sentences in the main text to coordinate the contents with SI and increase readability.

We have revised the sentence in the main text to explain the Langmuir confined fluid isobar model and RBM trajectory conversion as follows,

“The concave-up trajectories can be described by the Langmuir isobar adsorption model (Eq. 10) along with the RBM frequency conversion to $q[T]$ detailed in **Supplementary Text S14: b.**”

We have also included the detailed derivation in the current **Supplementary Text S14: b.**

“The positive second derivative means that the trajectory can be described by the Langmuir isobar in Eq. S6-4. To quantitatively analyze fluid isobars, we convert measured RBM frequencies into fractional coverage (q) from Eq. S11-9.^{4,5}

For a single-walled CNT, the vapor state RBM frequency (ω_V) is

$$\omega_V^2 = \frac{1}{r^2} \cdot \frac{Eh}{\rho h(1 - \nu^2)} \quad \text{Eq. S14-1}$$

where r is the SWCNT radius, E is Young’s modulus, h is the SWCNT wall thickness, ρ is the mass density, and ν is the Poisson ratio.

The liquid state RBM frequency (ω_L) is defined as

$$\omega_L^2 = \omega_V^2 + \frac{\gamma}{\rho h(1 - \nu^2)} \quad \text{Eq. S14-2}$$

where γ is an area-normalized spring constant characterizing the van der Waals coupling between the carbon shell and the first shell of adsorbed molecules. γ is related to the second derivative of the potential between the two shells.

We define the area-normalized spring constant at fractional coverage q as $q\gamma$, Eq. S14-2 becomes:

$$\omega_{RBM}^2 = \omega_V^2 + \frac{q\gamma}{\rho h(1 - \nu^2)} \quad \text{Eq. S14-3}$$

where ω_{RBM} is the RBM frequency at fractional coverage q . Combining Langmuir isobar in Eq. S6-4 and Eq. S14-3, it follows the Eq. 10 of the main text:

$$q[T] = \frac{\omega_{RBM}^2 - \omega_V^2}{\omega_L^2 - \omega_V^2} = \frac{e^{\frac{\Delta H}{RT}} P K_o}{1 + e^{\frac{\Delta H}{RT}} P K_o} \quad \text{Eq. S14-4}$$

where $q[T]$ is the surface coverage, ω_V (at which $q = 0$) is the lowest observed RBM frequency, ω_L (at which $q = 1$) is the highest observed RBM frequency, K_o is equilibrium constant, ΔH is the heat of adsorption, and P is the fluid pressure.”

Minor issues

Comment 2-2: When discussing the lack of the initial T_{max} for CNT X (page 10 line 216) the authors suggest this may be due to relaxed tension due to an initial focus at high laser power. Could the authors be more specific?

Response 2-2: Thank you for requesting details and information. We applied a laser power that allowed for facile positioning of the laser on the CNT. Centering a CNT is done by monitoring the Raman scattering response of the CNT and, for practical purposes, calls for ~0.3 mW of power. At this power, the heating of the CNT X is already substantial, impeding the subsequent observation of the initial trajectory as observed on CNT F and CNT G. In contrast, the initial power used for CNT F and CNT G is 0.01 mW.

Comment 2-2 (a): For instance (a) providing the expected temperature increase and

Response 2-2 (a): Temperature rise is dictated by the precise geometry of the problem (suspended CNT length, position of the laser spot, spatial extent of the laser power), absorption coefficient, and thermal conductivity of the CNT. The absorption coefficient is dictated by the CNT chirality and is subject to corrections from inter-tube coupling. The model we developed in our previous work was for single-walled carbon nanotubes (SWNTs)⁶. The Raman experiments for CNT X spectra were done ~10 μm off of the slit edge. The slit width is 35 μm , so we were not exactly in the center of the suspended section. The expected temperature rises to impede observing the first trajectory would need to be on the order of ~100 K above room temperature.

Comment 2-2 (b): Why the focused beam power differs from the one adopted on the other samples?

Response 2-2 (b): The reason is purely practical: with a higher beam power, there is more signal, allowing for faster centering on a CNT at the expense of heating it during that process. At a lower

beam power, centering the laser on a CNT can take excessively long. The time spent is about a few minutes over half an hour.

We have revised the sentence in the main text to describe the details of the experimental condition as follows,

“CNT X does not exhibit the initially high T_{\max} in Fig. 3e, (v)(vi), however, this self-tension may be relaxed due to an initial focus at high laser power (> 0.3 mW). At this power, the heating of the CNT X is already substantial, impeding the subsequent observation of the initial trajectory as observed on CNT F and CNT G. In contrast, the initial power used for CNT F and CNT G is 0.01 mW.”

Comment 2-3 (a): In my opinion, when discussing previous literature results at the light of the new model, the discussion is somewhat too qualitative.

For instance, when discussing previous data from Kumar et al. (page 16, line 248-352) could the authors provide actual numbers to substantiate the statement: “Their attribution to a damping term is now quantitatively described and validated in this work.”

Response 2-3 (a): We appreciate you bringing attention to this potential misconnection in the main text and current **Supplementary Text S9** in the SI. We had quantitatively described the damping calculation on **lines 185 to 191** on **pages 9-10** in the main text and compared the damping term with literature values, as follows,

“In contrast, the magnitude and direction of the shift are consistent with previous estimates of the specific damping (b) for CNTs. The outer shell of CNT G (35,6) has a carbon atom mass of 7.2×10^{-15} kg m^{-1} and exhibits an estimated frequency damping of $(\omega_{\text{RBM,max}})^2 - (\omega_{\text{RBM,min}})^2 = 1,969$ cm^{-2} or 44.4 cm^{-1} . This predicts 9.5 mN s m^{-2} as the specific damping change, in close agreement with the value of 8.3 mN s m^{-2} calculated by Kumar *et al.*⁷ for smaller diameter SWNT. ”

More detailed information is in **Supplementary Text S9: The Magnitude of the RBM Trajectories is Consistent with Damping.**

We further indicate this analysis and discussion in the revised sentence of the main text,

“Their attribution to a damping term is now quantitatively described and validated in this work. As discussed above, the damping coefficient of CNT G is 9.5 mN s m^{-2} , and more details are provided in **Supplementary Text S9.**”

Comment 2-3 (b): Alike, when discussing the contribution of multiple springs (Eq. 9), the authors state: “The smaller shift is consistent with some combination of a larger spring constant 373 and decreased damping from the fluid compared to the graphitic ribbon”. Could the authors elaborate further providing some figures?

Response 2-3 (b): We thank the reviewer for the comment regarding the contribution of multiple spring and dashpot combinations. **Eq. 9** predicts that mass couplings do not shift additively. The RBM frequency shift will be dominated by the largest magnitude terms in the force balance. The schematic illustration is provided within the **Fig. 5e** and **Fig. R2**. Our future study will describe this combination effect quantitatively.

$$\sum_{j=1}^N \left(\frac{\omega_{1,j}^2 \omega_{2,j}^2}{\omega_{1,j}^2 + \omega_{2,j}^2} \right) w[t] + \sum_{j=1}^N \left(\frac{\omega_{1,j}^2}{\omega_{1,j}^2 + \omega_{2,j}^2} \right) 4bw'[t] + w''[t] = 0 \quad \text{Eq. 9}$$

We have clarified this section as follows,

“**Eq. 9** above indicates that distinct mass couplings (graphitic ribbons plus interior water) do not shift additively; rather, the strongest coupling (i.e. interior water) dominates the RBM shift. Future work will describe this combination coupling effect quantitatively.”

Comment 2-4: Poor figure readability. Figures are generally too small to be readable. This is especially true for:

Fig. 1 panel (b), (d) and f (vi);

Fig. 2 first column (diffraction patterns)

Response 2-4: We thank the reviewer for the comments and suggestions. We have improved the readability and clarity of all **Fig. 1-5** in the main text.

For example, the revised **Fig. 1** and **Fig. 2** are shown below,

Fig. 1 | Platform and observations for the temperature-dependent Radial Breathing Mode (RBM) shift for isolated and suspended Double Walled Carbon Nanotubes (DWNTs).

Fig. 2 | Repeated heating and cooling scans for a series of DWNTs.

Comment 2-5: Eq. S5-1 is inconsistent with Eq S5-8 (the square root) and, both, seem inconsistent with S6-2 (β_2 is defined as the frequency on page 23 not frequency square).

Response 2-5: The linear expression presented in **Eq. S5-1 (current Eq. S6-1)** serves the purpose of providing a straightforward mathematical proof, which does not change the main conclusion regarding the falsification of fluid adsorption and the necessity of incorporating a damping term for the system of unopened and free-standing CNTs. To mitigate potential confusion arising from symbol representation, we further introduce symbols β_2 and $\gamma_2[T]$ in **Eq. S5-1 (current Eq. S6-1)**.

We revised the sentence to clarify the description of **Eq. S5-1 (current Eq. S6-1)** as follows,

“Describing the RBM frequency using a temperature (T) dependent spring constant imposed by an adsorbed fluid ($\gamma_2[T]$) with a linear expression for the purpose of a simple mathematical and conceptual elucidation:

$$\omega_{\text{RBM}}[T] = \beta_2 + \gamma_2[T] \quad \text{Eq. S6-1}$$

Here, β_2 is the RBM frequency in the absence of any environmental coupling (intrinsic spring constant of CNT).”

We also thank the reviewer for indicating the discrepancy between **Eq. S5-8 (current Eq. S6-8)** and **Eq. S6-2 (current Eq. S8-2)**

$$\omega_{\text{RBM}}[T] = \sqrt{\beta + \gamma[T] - b[T]^2} \quad \text{Eq. S6-8}$$

We have revised Eq. S5-8 (current Eq. S6-8) to maintain consistency with the expression for $\omega_{\text{RBM}}[T]$ as below and edited this section accordingly.

$$\omega_{\text{RBM}}[T] = \sqrt{\frac{\beta}{r^2} + \gamma[T] - b[T]^2} \quad \text{Revised Eq. S6-8}$$

$$\omega_{\text{RBM}}[T] = \sqrt{\omega_0^2 - b^2} \quad \text{Eq. S8-2}$$

where b is the damping coefficient, and $\omega_0^2 = \frac{\beta}{r^2} + \gamma$ is the sum of the intrinsic CNT and environmental coupling spring constants.

Comment 2-6: More of a scientific curiosity for future studies. An alternative technique to address vibration damping in nanostructures is time-resolved optical spectroscopy. With this technique, the breathing mode is directly excited by a pump beam, and the resulting trace allows us to directly visualize the displacement oscillation and damping in the time-domain. This technique has been applied to investigate individual nanostructures too. Are the authors aware of similar measurements carried out on single suspended CNT to retrieve the RBM frequency and damping vs temperature? If yes, it might be worth mentioning the reference to trigger the interest of the ultrafast community.

Response 2-6: Thank you for your insightful suggestion regarding alternative techniques for studying vibration damping in nanostructures. We agree with this comment and greatly appreciate the information provided by the reviewer. We appreciate its potential value for our future investigations, such as nanostructure dynamics, electron transfer, molecular vibrations, and structural changes within the nanoconfinement in ultrafast processes.

Comment 2-7: Summarizing, for the above-mentioned reasons I recommend Major revisions.

Response 2-7: We greatly thank and acknowledge all insightful comments that have improved and strengthened the quality, readability, and integrity of this manuscript. We hope the reviewer finds these comments well addressed by our revision and additional analysis presented above.

Reviewer #3 (Remarks to the Author):

Overall Comment: The paper by Tu *et al.* presents Raman experiments on free-standing individual carbon nanotubes (CNTs) assigned by electron diffraction. Reversible RBM shifts with temperature are measured in vacuum and assigned to interactions with the environment (adsorbed carbon species). The first study presents 3 different DWCNTs first (Fig. 1-4) with characteristic red (down) shifting RBM with temperature in vacuum (10⁻³-10⁻⁸ bar). The RBM shifts have hyperbolic shapes up to a max temperature of ~350-400K, after which the RBMs remain constant. This behavior is ascribed to vibrational damping from graphitic adsorbates located on the CNT external walls. Using a damped oscillator model, the authors argue that the RBM shifts are due to the dynamic force balance in the radial direction and not from additional mass onto the oscillator. While I appreciate the efforts behind these experiments and phenomenological model derivations, I am not sure the results represent a significant advance to the topic of nanotube sensing of confined fluids. RBM shifts by adsorbates such as water adsorption onto the inner or outer walls have been extensively studied in the past and analysis based on a damping harmonic oscillator has also been done (e.g. in Ref. 9). The shifting trend described here from carbon contaminants at the surface of DWCNTs is probably original and the model makes sense, but the significance of this work is unclear. I do not recommend publication in Nature Communication. This work is of interest for a more specialized journal, but there are major issues that will need to be addressed, as detailed below.

Response: We thank the reviewer for recognizing the significance of our work and for providing detailed comments. The delineation of two distinct mechanisms — the damped oscillator model from the graphitic carbon and the spring-dominant Langmuir confined fluid isobar model from the internal fluid — are newly described and validated by the nanotube systems investigated in this work. These observations and mechanisms would contribute to broad prospects in several fields, including nanosensor designs, applied physics, nanoscience, and nanomechanical systems. Your insightful comments have prompted us to include elucidating details and analysis in the manuscript to address the concerns.

Comment 3-1: Many Raman spectra involving different nanotubes and different temperatures were acquired. Except for Figure 1d, these Raman spectra are not shown, not even in the SI file. The only sequence in Figure 1d is very small and of low digital quality, which is problematic. Original spectra and fits to the experimental results should be presented more clearly.

Response 3-1: We thank the reviewer for requesting the full Raman spectra. We now include full spectra for each CNT studied in this work (CNT X, G, F, and the CNT in Fig. 5). These are in a new section of the SI, **Supplementary Text S2** and **Fig. S14-2**. We attach the revised **Fig. 1d** and the full Raman spectra of temperature scans of CNT F as an example (**Fig. R4**).

Revised Fig. 1d | Example Raman spectra of the as-grown, free-standing (19,3)@(22,11) DWNT as a function of local temperature heated by the excitation spot at 633 nm using a vacuum stage at $7.4 \cdot 10^{-7}$ bar. The Raman G-band was separately calibrated and used as a local thermometer of the CNT. Lorentzian fits (shaded curves) served to extract Raman mode frequencies.

Fig. R 3] 39 Raman spectra of CNT F scanned with a pressure of 7.4×10^{-4} mbar

“File: Matthias_CNT-F_210519_210518As01x1132_LaserPowerScan_633nm_p7.4e-4mbar”. Raman spectra of the first local laser power cycle scans (13 Raman spectra): **a**, RBM region and **b**, G band region. The full Raman scans of CNT F in this experiment: **c**, RBM region, and **d**, G band region.

Comment 3-2: Based on Figure 1d, RBMs are sharp at low T and more difficult to see at higher T, which makes me wonder how accurate the fits are to extract the positions and FWHMs for each run. Furthermore, the loss of intensity could also be due to a general loss of resonance, and this must be discussed somehow.

Response 3-2: We thank you for your question. From **Fig. 1d**, and in all of the data, the opposite is true. The features at moderate to high temperatures are sharp and clearly visible. The RBM broadens and weakens in intensity at room temperature. Note that the RBMs are still easily discernable and measurable in the room temperature spectra, and we provide examples in **Fig. 1d** and **Supplementary Text S2 (SI)**. Also, note that the major analysis of this work involves the higher temperature spectra, including the measurement of characteristic temperature (T_{max}), the minimum RBM frequency ($\omega_{RBM,min}$), and the maximum damping (b_{max}). Therefore, the parts of the data that are of the highest precision are used for these calculations.

Comment 3-3: Considering that the inner wall of a DWCNT is closely coupled to the outer wall, an absence of discussion of the inner wall RBM is surprising.

Response 3-3: We included an analysis of the two-shell DWNT system in **Supplementary Text S13** of the original SI (and now **Supplementary Text S7**). This analysis demonstrates that the conclusions remain unchanged when considering the more complex coupled shell oscillators. The reason for treating the system as a single shell in the main text analysis is that the RBM trajectory can be accurately described with a simple, understandable equation, highlighting the new physics involved (mechanical damping).

We now emphasize this point in the main text, inserting it after **Eq 2**.

“This **Eq 2** models the two shells of the DWNT as a single composite shell with a radial displacement $w[t]$ for simplicity. The more complex two-shell system is considered in **Supplementary Text S7**, where we show that it does not change the conclusions, with the error less than 2 cm^{-1} .”

Comment 3-4: Moreover, it is unclear why there is a need to ED assign the DWCNTs here. This information is not used anywhere in the analysis provided (qualitative comparisons with literature values are made in the SI file but not used).

Response 3-4: The reviewer may have missed that we, in fact, used the electron diffraction (ED) assigned DWCNT chirality in several places in the original manuscript, which enhances the precision of this work. Text on **lines 185 to 191** in the revised main text (**pages 9-10**) and all of current **Supplementary Text S9** in the SI incorporate the chirality of the shell to calculate the damping term. Additionally, the reviewer is correct in that we also use the chirality in current **Supplementary Text S13** to compare with literature values. In all, we feel that this work is strengthened by the inclusion of the ED shell assignments. Consequently, it is not clear what edit is being proposed here. Our inclination is to keep these valuable data in the manuscript.

Comment 3-5: The base pressure for those experiments is rather poor, in mbar range, and far from UHV. Hence, I am not sure the authors can safely rule out gas adsorption at the nanotube surface in those conditions.

Response 3-5: We would like to correct the reviewer that the measurements for most Raman scans in this work are in the nano-bar range (10^{-8} bar). In the sections, “**A New Understanding of One-Dimensional Vibrational Coupling**” in the main text and current **Supplementary Text S6** show that an adsorbing gas or fluid must have a positive second derivative in surface coverage with increasing temperature. Our observations for the unopened CNT systems in vacuum are distinctly negative or concave down (see **Fig. 2**); hence, we rule out an adsorbed gas phase. In support of this, in the text on **pages 18-20**, where the CNTs are opened and filled with water, we observed an RBM trajectory that is predicted by theory and concave up. On this basis, we formulated the two distinct mechanisms advanced in this work.

We do have experiments conducted in ultra-high vacuum (UHV) in the form of TEM imaging. We observed the dynamics of graphitic ribbons on the exterior surface (current **Fig. S5-2, Video S1**, and **Supplementary Text S5: b**) and the presence of a water-carbon oxidation product in the interior of CNTs (**Fig. 5d, Video S2**, and current **Supplementary Text S13: a**). These observations are consistent with our findings from Raman experiments and the mechanisms we formulated in this work.

Comment 3-6: Experiments on clean DWCNTs (exempt of carbonaceous contaminants at the surface) are also missing. This is needed to support the conclusion of a damping effect by graphitic ribbons.

Response 3-6: We thank the reviewer’s comment and information. However, we are not convinced that it is possible to produce suspended CNTs of this length free of carbon on the surface, and evidence appears to support that all measurements in the literature were on CNTs containing such carbon. We base this on the analysis that we conducted in current **Supplementary Text S13**, “**Comparison with Prior Literature Measurements of the RBM**”. The main point is that we measure the same RBM values as those published in the previous studies, such as those of Liu *et al.*⁸ (see current **Figure S13-1**). Therefore, the evidence suggests that those CNTs had the same carbon coupling and damping as the ones in this work. We conclude that it is not possible to generate CNTs of this type using a synthesis that necessarily produces carbon fragments and also keeps such fragments from van der Waal adhesion to the sidewall. Furthermore, we are not sure how one could prove that a CNT was completely devoid of carbon fragments and short of TEM imaging along the entire length of each CNT studied. We did attempt to bake out the system, and the results are reported in **Fig. 4e**, further suggesting that such carbon cannot be thermally removed.

Comment 3-7: Equation 5 to evaluate T_{\max} deals with length units that can be problematic. The dimensions of the nanotubes are rather small, and this may give speculative estimates of T_{\max} . Please explain more clearly the choices of length scales that were used to reach the 580K value.

Response 3-7:

We thank you for your comment and concern regarding **Eq. 5**. These calculations are in the main text of the manuscript. The value of $M = 0.5$ GPa is Young's modulus⁹ for the CNT, while $S = 2$ μm is the size of the Raman spot used in this work. The value of α is the coefficient of thermal expansion (CTE) and has been measured and reported recently. The tension value T_{n_0} was estimated as described on **page 11**: "Bunch *et al.* calculated a tension of 13 nN for a 2 μm wide graphene resonator¹⁰. Scaled to a 3 nm DWNT, the initial 33 pN tension (T_{n_0}) translates into an axial lattice strain of 0.61% at T_0 using a Young's modulus (M) of 0.5 GPa⁹." The **Eq. 5** includes only values taken from the literature, and no length units are used in the calculations. The agreement between the predicted and measured T_{max} values support the mechanism advanced in this work.

$$T_{\text{max}} = \frac{T_{n_0}}{MS\alpha} + T_0 \quad \text{Eq. 5}$$

Comment 3-8: The discussion on maximum damped limits (p. 12-14) and FWHM behavior (p. 15 and Fig. 4f) is very confusing and hard to follow. Quantitative results from actual spectra are not shown and it is hard to understand how inhomogeneity along the nanotubes can contribute to changes in FWHM or damping forces. The discussion related to the homogeneity and broadening with the number of carbonaceous tethers at T_0 vs. T_{max} is unclear in the current form of the paper. Further experimental evidence is probably needed to support these claims.

Response 3-8: We thank you for your insightful comments and questions, and we have clarified these sections. The central point of this section is the observation that the spring and damping components of the RBM appear to vary together linearly. The linearity is obvious from **Fig. 4a-c** and **Fig. 4e**. It is described by **Eq. 7** and **Eq. 8**, which explain the dependence. In short, each carbon fragment brings a spring but also a damping component. As the density of carbon fragments n changes, the frequency is predicted to remain invariant as we observe. A similar relation appears to hold of the inhomogeneous broadening leading to the FWHM.

Additional quantitative spectra are now included in response to Reviewer 3's first comment (**Supplementary Text S2**) and should address the request for more spectra here.

We have revised the text in these sections to be clearer.

"The RBM reaches its minimum frequency ($\omega_{\text{RBM,min}} = \sqrt{\omega_0^2 - b_{\text{max}}^2}$) at zero tension for all $T > T_{\text{max}}$, and the reproducibility of this limit is striking as different sections of the same CNT are scanned. We plotted the square of the maximum damping (b_{max}^2) versus the square of the net spring constant (ω_0^2) for every scan of each DWNT in this work (**Fig. 4a-c**). The data disperse linearly, especially for CNT X and CNT F (**Fig. 4b** and **c**), suggesting:

$$b_{\text{max}}^2 = \nu\omega_0^2 + l \quad \text{Eq. 7}$$

where ν is the dimensionless slope ($\text{cm}^{-2}/\text{cm}^{-2}$) with intercept l ."

Comment 3-9: RBM shifts with temperature in Figure 5 due to water confinement inside nanotubes is an extension of past work in Ref. 24. It is not clear what CNTs are studied here, and again no Raman spectra are presented.

Response 3-9: We now provide additional spectra for this section as well in a new section of **Supplementary Text S2** and **Fig. S14-2**. We discuss the current work in relation to Ref. 24 from 2016¹¹ on **pages 20-21** of the manuscript. First, we point out that “The RBM frequency measurements in this current work are more precise at 0.2 cm⁻¹ resolution...” Furthermore, the section describes that the resolution is improved enough to apply a quantitative theory, the Langmuir isobar adsorption model, to the data, as we have done with **Eq. 10 and Eq. 11**. We think that the readers are certain to identify a major advance over the previous Ref 24 study.

We have revised this section to specify the more precise Raman resolution and elucidate the fluid phase transition tracing the Langmuir isobar model.

“The RBM frequency measurements in this current work are more precise at 0.2 cm⁻¹ resolution, and the isobars measured in **Fig. 5** exhibit a more nuanced picture of the fluid phase transition, which occurs over a broad window of temperatures. Instead of a sharp boundary, the fluid occupancy decays exponentially following a quantitative theory, the Langmuir isobar adsorption model (**Eq. 10 and Eq. 11**).”

Comment 3-10: Moreover, I am not sure I understand the relation with the other experiments in Figure 1-4. What new knowledge was gain here compared to past studies on water is unclear. Is this the topic of another paper or simply an example of application of the damped oscillator model?

Response 3-10: Both the strain-dependent and the water-filling experiments involve mass coupling to the sidewall of CNTs, and both shift the RBM frequency with temperature. However, the trajectories that we measure differ significantly and, in a way, are predicted by theory (refer to current **Supplementary Text S6** and the discussion in the main text.). Additionally, our study provides new insights into water behavior, including the observation and mathematical description of the fluid isobar — a way to measure the thermodynamic properties of water under confinement. The water-filled CNTs produced using the procedure on the suspended system of this current work also apparently exhibit an O-H vibrational mode detected in our study using Electron Energy Loss Spectroscopy (EELS) characteristic of confined water. This aspect of our study is currently under review in the Nature Journal and *arXiv paper*. We will explore this phenomenon further in future work.

Reviewer #4 (Remarks to the Author):

Overall Comment: The manuscript presents a comprehensive investigation into the Radial Breathing Mode (RBM) frequencies exhibited by isolated suspended double-walled carbon nanotubes when coupled with external environments and confined fluid within their inner cavities. The study utilizes transmission electron microscopy, electron diffraction, and micro-Raman spectroscopy with three laser excitations to analyze a series of nanotubes for statistical purposes. Additionally, supplementary information detailing the Raman setup and the precise protocol for recording the RBM frequencies is provided. A proposed model, integrating a harmonic oscillator and damping system, effectively explains the observed phenomena. Moreover, the paper is expected to appeal to a wider audience within the community interested in investigating filled carbon nanotubes. Notably, the mechanical properties of nanotubes may be influenced not only by solid species adsorbed on their outer surface but also by those encapsulated within their interior (molecules, carbon allotropes, crystals), potentially resulting in significant modifications in the RBM frequencies, including their cancellation. Overall, this is a very good manuscript.

With some enhancements, I recommend the publication of this manuscript.

Response: We are grateful for the reviewer's positive feedback. We are truly excited that you acknowledge the experimental designs, damping oscillator mechanisms, the discovery of confined fluid isobars, and the broader implications of our work.

Comment 4-1: The abstract could be less technical or simplified for a broader audience.

Response 4-1: We thank the reviewer for your suggestions. We have simplified the abstract as a clear language as follows,

The revised abstract:

“Because of their large surface areas, nanotubes and nanowires demonstrate exquisite mechanical coupling to their surroundings, promising advanced sensors and nanomechanical devices. However, this environmental sensitivity has resulted in several ambiguous observations of vibrational coupling across various experiments. Herein, we demonstrate a temperature-dependent Radial Breathing Mode (RBM) frequency in free-standing, electron-diffraction-assigned Double-Walled Carbon Nanotubes (DWNTs) that shows an unexpected and thermally reversible frequency downshift of 10 to 15%, for systems isolated in vacuum. An analysis based on a harmonic oscillator model assigns the distinctive frequency cusp, produced over 93 scans of 3 distinct DWNTs, along with the hyperbolic trajectory, to a reversible increase in damping from graphitic ribbons on the exterior surface. Strain-dependent coupling from self-tensioned, suspended DWNTs maintains the ratio of spring-to-damping frequencies, producing a stable saturation of RBM in the low-tension limit. In contrast, when the interior of DWNTs is subjected to a water-filling process, the RBM thermal trajectory is altered to that of a Langmuir isobar and elliptical trajectories, allowing measurement of the enthalpy of confined fluid phase change. These mechanisms and quantitative theory provide new insights into the environmental coupling of nanomechanical systems and the implications for devices and nanofluidic conduits.”

Comment 4-2: Regarding line 61, perhaps rephrase to avoid using 'absent' as follows: 'In the absence of an underlying substrate...'

Response 4-2: Thank you for your suggestion.

The revised sentence is provided below,

“The platform consists of an isolated, ultralong (mm) DWNT grown across 13 slits (most of which are 35 μm) of a 3 mm in diameter homemade TEM chip such that micro-Raman spectroscopy can be performed on the suspended section in the absence of an underlying substrate (**Fig. 1a, b**).”

Comment 4-3: Additionally, in line 167, Equation 2 is referenced without prior presentation.

Response 4-3: The radial deflection $w[t]$ has been applied to describe radial breathing mode in several literatures¹²⁻¹⁵. The damping coefficient and mechanism have been mentioned in Kumar *et al.*⁷

We had included the references in the original main text, as in the revised sentences below.

“The RBM is described using a harmonic oscillator model with the radial force balance representing the radial displacement $w[t]$ containing terms accounting for the intrinsic CNT spring constant contribution ($\frac{\beta}{r^2}$), where r is the CNT shell radius and β is the intrinsic spring constant, and environmental contribution (γ) (**Fig. 3a**)¹²⁻¹⁵.”

“A damping term proportional to coefficient ($b = \frac{\bar{b}}{2\rho}$) has been proposed⁷ but not yet used quantitatively to describe the RBM (**Supplementary Text S6 and Fig. S6-1**).”

$$\left(\frac{\beta}{r^2} + \gamma\right)w[t] + \frac{\bar{b}}{2\rho}w'[t] + w''[t] = \omega_o^2w[t] + 2bw'[t] + w''[t] = 0 \quad \text{Eq. 2}$$

Comment 4-4: In the caption of Figure 4, it would be beneficial to define FWHM (Full Width at Half Maximum) and specify what it pertains to.

Response 4-4: We thank the reviewer for this comment and apologize for this omission. The Full width at half maximum (FWHM) in this work is the difference between the two RBM frequencies at which the intensity equals half of its maximum value. We have included this information in the main text.

The revised sentence is as follows,

“The full width at half maximum (FWHM) of the RBM peak, which is the difference between the two RBM frequencies at which the intensity equals half of its maximum value, is broadened at room temperature, when b is inhomogeneous and narrow at high temperature, when a uniform, limiting b_{\max} is reached, consistent with an inhomogeneous broadening mechanism.”

Comment 4-5: Finally, could this model be applicable to single-walled carbon nanotubes and their environmental interactions?

Response 4-5: Yes, our models in this manuscript describe two shells of the DWNT as a single composite shell with a radial displacement $w[t]$ for simplicity, as shown in **Eq. 2**. Also, in response to Reviewer 3's comment, we emphasize an analysis of the two-shell DWNT system in **Supplementary Text S13** of the original **SI** (and now **Supplementary Text S7**). This analysis demonstrates that there is no change in the conclusions if one considers the more complex coupled shell oscillators.

Responses to the concluding comments: We greatly appreciate the reviewers' acknowledgment of the novelty and the significance of our work, as well as their recognition of the prospective features of the proposed damped oscillator model and Langmuir isobar model for the broad impacts. Additionally, we extend our deep gratitude for all insightful comments that have improved and strengthened the quality and integrity of this manuscript. We believe that our responses and additional information presented above adequately address these comments. We anticipate the editor and reviewers' agreement. We look forward to contributing our findings to the **Nature Communications** journal.

Reference:

- 1 Tournus, F., Latil, S., Heggie, M. & Charlier, J.-C. π -stacking interaction between carbon nanotubes and organic molecules. *Phys. Rev. B* **72**, 075431 (2005).
- 2 Georgakilas, V., Perman, J. A., Tucek, J. & Zboril, R. Broad family of carbon nanoallotropes: classification, chemistry, and applications of fullerenes, carbon dots, nanotubes, graphene, nanodiamonds, and combined superstructures. *Chem. Rev.* **115**, 4744-4822 (2015).
- 3 Kitt, A. L. *et al.* How graphene slides: measurement and theory of strain-dependent frictional forces between graphene and SiO₂. *Nano Lett.* **13**, 2605-2610 (2013).
- 4 Longhurst, M. J. & Quirke, N. The environmental effect on the radial breathing mode of carbon nanotubes in water. *J. Chem. Phys.* **124**, 234708, doi:10.1063/1.2205852 (2006).
- 5 Longhurst, M. J. & Quirke, N. The environmental effect on the radial breathing mode of carbon nanotubes. II. Shell model approximation for internally and externally adsorbed fluids. *J. Chem. Phys.* **125**, 184705, doi:10.1063/1.2360943 (2006).
- 6 Kuehne, M. *et al.* Impedance of Thermal Conduction from Nanoconfined Water in Carbon Nanotube Single-Digit Nanopores. *J. Phys. Chem. C* **125**, 25717-25728, doi:10.1021/acs.jpcc.1c08146 (2021).
- 7 Kumar, R., Aykol, M. & Cronin, S. B. Effect of nanotube-nanotube coupling on the radial breathing mode of carbon nanotubes. *Phys. Rev. B* **78**, 165428 (2008).
- 8 Liu, K. *et al.* Quantum-coupled radial-breathing oscillations in double-walled carbon nanotubes. *Nat. Commun.* **4**, 1375 (2013).
- 9 Zhang, C. *et al.* Mechanical properties of carbon nanotube fibers at extreme temperatures. *Nanoscale* **11**, 4585-4590 (2019).
- 10 Bunch, J. S. *et al.* Electromechanical resonators from graphene sheets. *Science* **315**, 490-493 (2007).
- 11 Agrawal, K. V., Shimizu, S., Drahushuk, L. W., Kilcoyne, D. & Strano, M. S. Observation of extreme phase transition temperatures of water confined inside isolated carbon nanotubes. *Nat. Nanotechnol.* **12**, 267-273 (2017).
- 12 Longhurst, M. & Quirke, N. The environmental effect on the radial breathing mode of carbon nanotubes in water. *J. Chem. Phys.* **124**, 234708 (2006).
- 13 Wang, C., Ru, C. & Mioduchowski, A. Applicability and limitations of simplified elastic shell equations for carbon nanotubes. *J. Appl. Mech.* **71**, 622-631 (2004).
- 14 Longhurst, M. & Quirke, N. The environmental effect on the radial breathing mode of carbon nanotubes. II. Shell model approximation for internally and externally adsorbed fluids. *J. Chem. Phys.* **125**, 184705 (2006).
- 15 Longhurst, M. & Quirke, N. Pressure dependence of the radial breathing mode of carbon nanotubes: the effect of fluid adsorption. *Phys. Rev. Lett.* **98**, 145503 (2007).

REVIEWERS' COMMENTS

Reviewer #1 (Remarks to the Author):

The authors have addressed all my previous comments. The manuscript can be accepted for publication in the present form.

Reviewer #2 (Remarks to the Author):

I reviewed the manuscript "Environmental Damping and Vibrational Coupling of Confined Fluids within Isolated Carbon Nanotubes" together with SI. My concerns have been substantially addressed in the revised version. I therefore recommend acceptance of the manuscript in its present form.

Reviewer #3 (Remarks to the Author):

My concerns have been addressed in this revised manuscript. I recommend publication in Nature Communications without further review.

Reviewer #4 (Remarks to the Author):

The authors of this manuscript have adequately addressed all the critiques and remarks from the referees. Significant changes have been made regarding the model and the interpretation of the experimental results. Therefore, I recommend the publication of the manuscript in Nature Communications with a minor correction: Equation 10 should be introduced after the sentence '... detailed in Supplementary Text S14: b:'.

RESPONSE TO REVIEWERS' COMMENTS

Reviewer #1 (Remarks to the Author):

Overall Comment: The authors have addressed all my previous comments. The manuscript can be accepted for publication in the present form.

Response: We thank you for your positive feedback and agreement on our work.

Reviewer #2 (Remarks to the Author):

Overall Comment: I reviewed the manuscript “Environmental Damping and Vibrational Coupling of Confined Fluids within Isolated Carbon Nanotubes” together with SI. The revised version substantially addresses my concerns. Therefore, I recommend accepting the manuscript in its present form.

Response: We greatly appreciate the reviewer’s time in reviewing our main text and SI. We are very pleased with your recommendation to accept our manuscript.

Reviewer #3 (Remarks to the Author):

Overall Comment: My concerns have been addressed in this revised manuscript. I recommend publication in Nature Communications without further review.

Response: We thank you for your insightful and positive comments on our manuscript. We appreciate your recommendation for the publication of our work.

Reviewer #4 (Remarks to the Author):

Overall Comment: The authors of this manuscript have adequately addressed all the critiques and remarks from the referees. Significant changes have been made regarding the model and the interpretation of the experimental results. Therefore, I recommend the publication of the manuscript in Nature Communications with a minor correction: Equation 10 should be introduced after the sentence ‘... detailed in Supplementary Text S14: b:’

Response: We greatly appreciate the reviewer’s remarks on the significant changes in the model and experimental results of our revised manuscript. We thank you for your suggestion for a minor correction and your recommendation for the publication of our manuscript.

The revised sentence is provided below,

“The concave-up trajectories can be described by the Langmuir isobar adsorption model along with the RBM conversion to $q[T]$ detailed in **Supplementary Text S14: b and Eq. (10).**”